# Benchmarking Stochastic Approximation Algorithms for Fairness-Constrained Training of Deep Neural Networks

**Andrii Kliachkin**,\* **Jana Lepšová**\*, **Gilles Bareilles**\* & **Jakub Marecek**
Artificial Intelligence Center,
Czech Technical University in Prague
`firstname.lastname@fel.cvut.cz`

## Abstract

The ability to train Deep Neural Networks (DNNs) with constraints is instrumental in improving the fairness of modern machine-learning models. Many algorithms have been analysed in recent years, and yet there is no standard, widely accepted method for the constrained training of DNNs. In this paper, we provide a challenging benchmark of real-world large-scale fairness-constrained learning tasks, built on top of the US Census (Folktables, Ding et al. (2021)). We point out the theoretical challenges of such tasks and review the main approaches in stochastic approximation algorithms. Finally, we demonstrate the use of the benchmark by implementing and comparing three recently proposed, but as-of-yet unimplemented, algorithms both in terms of optimization performance, and fairness improvement. We release the code of the benchmark as a Python package at `https://github.com/humancompatible/train`.

## 1 Introduction

There has been a considerable interest in detecting and mitigating bias in artificial intelligence (AI) systems, recently. Multiple legislative frameworks, including the AI Act in the European Union, require the bias to be removed, but there is no agreement on what the correct definition of bias is or how to remove it. A natural translation of the requirement of removing bias into the formulation of training of deep neural network (DNN) utilizes constraints bounding the difference in empirical risk across multiple subgroups (Chen et al., 2018; Nandwani et al., 2019; Ravi et al., 2019). Over the past five years, there have been numerous algorithms proposed to solve convex and non-convex empirical-risk minimization (ERM) problems subject to constraints bounding the absolute value of empirical risk (Fang et al., 2024; Berahas et al., 2021a; Curtis et al., 2024a; Oztoprak et al., 2023; Berahas et al., 2021b; Na et al., 2023a;b; Bollapragada et al., 2023; Curtis et al., 2024b; Shi et al., 2022; Facchinei & Kungurtsev, 2023; Huang et al., 2025; Huang & Lin, 2023). Numerous other algorithms of this kind could be construed, based on a number of design choices, including:

- sampling techniques for the ERM objective and the constraints, either the same or different;
- use of first-order or higher-order derivatives, possibly in quasi-Newton methods;
- use of globalization strategies such as filters or line search;
- use of "true" globalization strategies including random initial points and random restarts in order to reach global minimizers.

Nevertheless, there is no single toolkit implementing the algorithms, which would allow for their easy comparison, and there is no benchmark to test the combinations of design choices on.

In this paper, we consider the constrained ERM problem:

$$\min_{x \in \mathbb{R}^n} \mathbb{E}[f(x, \xi)] \quad \text{s.t.} \quad \mathbb{E}[c(x, \zeta)] \leq 0, \tag{1}$$

where $\xi$ and $\zeta$ are random variables. Further, we provide an automated way of constructing the ERM formulations out of a computation graph of a neural network defined by PyTorch or TensorFlow, the

---

\*Equal contribution, more junior authors listed first.

Table 1: Particular formulations of the constraint function $c$ to enforce fairness.

| Model | Our formulation |
|---|---|
| Accuracy equality | $|\mathbb{E}_{\mathcal{D}[\text{group } A]}[\ell(f_\theta(X), Y)] - \mathbb{E}_{\mathcal{D}[\text{group } B]}[\ell(f_\theta(X), Y)]| \leq \delta$ |
| Equal opportunity Hardt et al. (2016) | $|\mathbb{E}_{\mathcal{D}[\text{group } A, Y=+]}[\ell(f_\theta(X), Y)] - \mathbb{E}_{\mathcal{D}[\text{group } B, Y=+]}[\ell(f_\theta(X), Y)]| \leq \delta$ |
| Equalized odds Hardt et al. (2016) | $\sum_{v \in \{+, -\}} |\mathbb{E}_{\mathcal{D}[\text{group } A, Y=v]}[\ell(f_\theta(X), Y)] - \mathbb{E}_{\mathcal{D}[\text{group } B, Y=v]}[\ell(f_\theta(X), Y)]| \leq \delta$ |

choice of the constraints (see Table 1), and a definition of the protected subgroups to apply constraints to. Specifically, we provide means of utilizing the US Census data via the Python package Folktables, together with definitions of up to 5.7 billion protected subgroups. This presents a challenging benchmark in stochastic approximation for the constrained training of deep neural networks.

**Our contributions.** The contributions of this paper are:

- a literature review of algorithms subject to handling (1);
- a toolbox that *(i)* implements four algorithms applicable in real-world situations, and *(ii)* provides an easy-to-use benchmark on real-world fairness problems;
- numerical experiments that compare these algorithms on a real-world dataset, and a comparison with alternative approaches to fairness.

**Paper structure.** The rest of the paper is organized as follows. Section 2 reviews related works and presents the relevant notions of fairness. Section 3 introduces the algorithms. Section 4 reports on our experiments. Section 5 concludes.

## 2 RELATED WORK, AND BACKGROUND IN FAIRNESS

In the literature on fairness, one distinguishes among pre-processing, in-processing, and post-processing. Pre-processing methods focus on modifying the training data to mitigate biases (Tawakuli & Engel, 2024; Du et al., 2021). In-processing methods enforce fairness during the training process by modifying the learning algorithm itself (Wan et al., 2023). Post-processing methods adjust the model's predictions after training (Kim et al., 2019). The constrained ERM approach (1) belongs to the class of in-processing methods.

In-processing methods include several approaches. One trend consists in jointly learning a predictor function and an adversarial agent that aims to reconstitute the subgroups from the predictor (Adel et al., 2019; Louppe et al., 2017; Madras et al., 2018; Edwards & Storkey, 2016). Another approach consists in adding "penalization" terms to the empirical risk term. These additional penalization terms, commonly referred to as regularizers, promote models that are a compromise between fitting the training data, and optimizing a fairness metric. Differentiable regularizers include, among others, HSIC (Li et al., 2022), Fairret (Buyl et al., 2024), or Prejudice Remover (Kamishima et al., 2012).

Closer to our setting, Cotter et al. (2019) consider minimizing the empirical risk subject to the so-called rate constraints based on the models prediction rates on different datasets. These rates, derived from a dataset, give rise to non-convex, non-smooth, and large-scale inequality constraints akin to (1). Cotter et al. (2019) argue that hard constraints, although leading to a more difficult optimization problem, offer advantages over using a weighted sum of multiple penalization terms. Indeed, while the choice of weights for the penalization terms may depend on the dataset, specifying one constraint for each goal is easier for practitioners. In addition, a penalization-based model provides a predictor that balances minimizing the data-fit term and penalties in an opaque way, whereas a constraint-based model allows for a clearer understanding of the model design: minimizing the data-fit term subject to "hard" fairness constraints. Rate constraints differ from those in (1) in that they are piecewise-constant, rendering first-order methods unsuitable for solving them. We refer to the recent work of Ramirez et al. (2025) for a detailed argument on why constraining ERM problems is preferable to penalizing the ERM with multiple terms.

Major toolboxes for evaluating the fairness of models or for training models with fairness guarantees include AIF360 (Bellamy et al., 2018) and FairLearn (Bird et al., 2020). Delaney et al. (2024) compute the Pareto front of accuracy and fairness metrics for high-capacity models, and Buyl et al.

(2024) provides differentiable fairness-inducing penalization terms. We also note the recent Cooper toolbox, closest to our setting, with Lagrangian-based methods focus (Gallego-Posada et al., 2025).

Le Quy et al. (2022) provides a detailed survey of fairness-oriented datasets, and Ding et al. (2021) derives new datasets. The benchmark of Han et al. (2023) reviews the existence of biases in prominent datasets, finding that "not all widely used fairness datasets stably exhibit fairness issues", and assesses the performance of a range of in-processing methods in addressing biases, focusing on differentiable minimization only. Other benchmarks of fairness methods include Defrance et al. (2024); Fabris et al. (2022); Pessach & Shmueli (2022); Chen et al. (2024). Statistical aspects of the fairness-constrained Empirical Risk Minimization have only been considered recently; see e.g. Chamon et al. (2022).

The template problem (1) encompasses fairness-enforcing approaches that find applications in high-risk domains, such as credit scoring, hiring processes, medicine and healthcare (Chen et al., 2023), ranking and recommendation (Pitoura et al., 2022), but also in forecasting the observations of linear dynamical systems (Zhou et al., 2023b), or in two-sided economic markets (Zhou et al., 2023a). In addition, solving (1) is of interest in other fields, such as compression of neural networks (Chen et al., 2018), improving statistical performance of neural networks (Nandwani et al., 2019; Ravi et al., 2019), or the training of neural networks with constraints on the Lipschitz bound (Pauli et al., 2021). We note that all the aforementioned methodologies feature large-scale constraints.

**Deep neural networks (DNNs).** Consider a dataset of $N$ observations $\mathcal{D} = \{(X_i, Y_i), i = 1, ..., N\}$. We seek some function $f_\theta$ such that $f_\theta(X_i) \approx Y_i$. A typical formulation of this task is the following regression problem:

$$\min_{\theta \in \mathbb{R}^n} \frac{1}{N} \sum_{i=1}^N \ell(f_\theta(X_i), Y_i) + \mathcal{R}(\theta). \tag{2}$$

Here, $\ell : \mathbb{R} \times \mathbb{R} \to \mathbb{R}$ is a loss function, such as the logistic loss $\ell(y; z) = \log(1 + e^{-yz})$, the hinge loss $\ell(y; z) = \max\{0, 1 - yz\}$, the absolute deviation loss $\ell(y; z) = |y - z|$, or the square loss $\ell(y; z) = \frac{1}{2}(y - z)^2$. The term $\mathcal{R}$ is a regularizer, and $f_\theta$ is a deep neural network (DNN) of depth $L$ with parameters $\theta$. The DNN $f_\theta$ is defined recursively, for some input $X$, as

$$a_0 = X, \qquad a_i = \rho_i(V_i(\theta)a_{i-1}), \text{ for every } i = 1, \ldots, L, \qquad f_\theta(X) = a_L, \tag{3}$$

where $V_i(\cdot)$ are linear maps into the space of matrices, and $\rho_i$ are activation functions applied coordinate-wise, such as ReLU $\max(0, t)$, quadratics $t^2$, hinge losses $\max\{0, t\}$, and SoftPlus $\log(1 + e^t)$. A dataset $\mathcal{D}$ is described by attributes (or features), such as age, income, gender, etc. The attribute which the DNN is trained to predict is called the class attribute. We denote the class attribute by $Y$, whereas the predicted value given by the DNN is denoted by $\hat{Y}$. Both $Y$ and $\hat{Y}$ are binary and take values in $\{+, -\}$.

**Fairness-aware learning applied to DNNs.** The goal of this approach is to reduce discriminatory behavior in the predictions of a DNN across different demographic groups (e.g., male vs. female). The demographic groups are also reffered to as subgroups. The attributes such as race or gender which must be handled cautiously are called protected. We denote by $S$ the protected attribute or more generally the set of groups defined by multiple protected attributes, $s_1, \ldots, s_m$ its possible values or the indicator of membership in the groups, and $\mathcal{D}[s_i]$ the observations in $\mathcal{D}$ such that $S = s_i$. A way to impose fairness on the learned predictor is to equip (2) with suitable constraints. Some possible constraint choices are shown in Table 1. Choosing loss difference bound as the constraint, denoting $\ell^{s_i}(\theta) = \frac{1}{|\mathcal{D}[s_i]|} \sum_{X, Y \in \mathcal{D}[s_i]} \ell(f_\theta(X), Y)$ for $i = 1, \ldots, m$, and setting $\delta > 0$ yields formulation:

$$\min_{\theta \in \mathbb{R}^n} \quad \frac{1}{N} \sum_{i=1}^N \ell(f_\theta(X_i), Y_i) + \mathcal{R}(\theta)$$
$$\text{s.t.} \quad -\delta \leq \ell^{s_i}(\theta) - \frac{1}{m} \sum_{j=1}^m \ell^{s_j}(\theta) \leq \delta, \quad i = 1, \ldots, m. \tag{4}$$

Bounding the distance between subgroup losses yields $m$ constraints. Other formulations are possible, such as bounding the distance between every pair of subgroup, providing simpler individual constraints, but in greater number ($m(m + 1)/2$). Formulation (4) extends to several protected

Table 2: Three elementary notions of fairness

| Independence | Separation | Sufficiency |
|---|---|---|
| $\frac{1}{m}\sum_{i=1}^{m}\left\|P_i^{\mathrm{ind}} - \frac{1}{m}\sum_j P_j^{\mathrm{ind}}\right\|$ | $\frac{1}{2}\sum_{v\in\{+,-\}}\frac{1}{m}\sum_{i=1}^{m}\left\|P_{i,v}^{\mathrm{Sp}} - \frac{1}{m}\sum_j P_{j,v}^{\mathrm{Sp}}\right\|$ | $\frac{1}{2}\sum_{v\in\{+,-\}}\frac{1}{m}\sum_{i=1}^{m}\left\|P_{i,v}^{\mathrm{Sf}} - \frac{1}{m}\sum_j P_{j,v}^{\mathrm{Sf}}\right\|$ |

attributes by adding the corresponding set of equations; we omit this direct generalisation for clarity. Note that we employ constraints based on the loss function, as it is a continuous function, amenable to nonsmooth nonconvex optimization. For constraints involving directly discontinuous quantities such as accuracies and rates, see Cotter et al. (2019).

**Fairness metrics.** There exist tens of fairness metrics (Verma & Rubin, 2018). However, Barocas et al. (2023, Ch. 3) pointed out that most fairness metrics are combinations of three elementary fairness criteria: independence, separation, and sufficiency. These criteria cannot be minimized simultaneously, and there is a trade-off between attaining the elementary fairness metrics and the prediction accuracy, i.e., the probability that the predicted value is equal to the actual value. Thus, we seek an optimal trade-off between attaining the fairness metrics and minimizing the prediction inaccuracy. We next recall the definitions of these three relevent metrics, following Barocas et al. (2023), and provide the formula for computing them in Table 2.

**Independence (Ind)** This fairness criterion requires the prediction $\hat{Y}$ to be statistically independent of the protected attribute $S$. Equivalent definitions of independence for a binary classifier $\hat{Y}$ are referred to as statistical parity (SP), demographic parity, and group fairness. Independence is the simplest criterion to work with, both mathematically and algorithmically. In a binary classification task, independence implies the equality of $P_i^{\mathrm{ind}} = P(\hat{Y} = + \mid S = s_i)$ for all $i = 1, \ldots, m$.

**Separation (Sp)** Unlike independence, the separation criterion requires the prediction $\hat{Y}$ to be statistically independent of the protected attribute $S$, given the true label $Y$. The separation criterion also appears under the name Equalized odds (EO). In a binary classification task, the separation criterion requires that all groups experience the same true negative rate and the same true positive rate. Formally, we require the equality of $P_{i,v}^{\mathrm{Sp}} = P(\hat{Y} = + \mid S = s_i, Y = v)$ for every $i = 1, \ldots, m$, and $v \in \{+, -\}$.

**Sufficiency (Sf)** The sufficiency criterion is satisfied if the true label $Y$ is statistically independent of the protected attribute $S$, given the prediction $\hat{Y}$. In a binary classification task, the sufficiency criterion requires a parity of positive and negative predictive values across the groups. Formally, we require the equality of $P_{v,s}^{\mathrm{Sf}} = P(Y = + \mid \hat{Y} = v, S = s)$, for every $i = 1, \ldots, m$, and $v \in \{+, -\}$.

## 3 ALGORITHMS

We recall that we consider the optimization problem

$$\min_{x\in\mathbb{R}^n} F(x) \quad \text{s.t.} \quad C(x) \leq 0, \tag{5}$$

where the functions $F : \mathbb{R}^n \to \mathbb{R}$ and $C : \mathbb{R}^n \to \mathbb{R}^m$ are defined as expectations of functions $f$ and $c$, which depend on random variables $\xi$ and $\zeta$, respectively. Solving (5) has the following challenges:

- large-scale objective and constraint functions, which require sampling schemes,
- the necessity of incorporating inequality constraints, not merely equality constraints (see fairness formulations in Table 1),
- the necessity to cope with the nonconvexity and nonsmoothness of $F$ and $C$, due to the presence of neural networks.

In this section, we identify the algorithms that address these challenges most precisely. However, we note that there exists currently no algorithm with guarantees for such a general setting.

**Recalls and notation.** We denote the projection of a point $x$ onto a set $\mathcal{X}$ by $\mathrm{proj}_{\mathcal{X}}(x) = \arg\min_{v\in\mathcal{X}}\|x - v\|^2$. We denote by $N \sim \mathcal{G}(p_0)$ sampling a random variable from the geometric distribution with a parameter $p_0$, i.e., the probability that $N = n$ equals $(1 - p_0)^n p_0$ for $n \geq 0$. We

Table 3: Assumptions on objective and constraint functions, $F$ and $C$, which allow for theoretical convergence proofs.

| | Objective function $F$ | | | | Constraint function $C$ | | | | | | |
| Algorithm | stochastic | weakly convex | $\mathcal{C}^1$ with Lipschitz $\nabla F$ | tame loc. Lipschitz | stochastic | $C(x)=0$ | $C(x)=0$ and $C(x)\leq 0$ | linear | weakly convex | $\mathcal{C}^1$ with Lipschitz $\nabla C$ | tame loc. Lipschitz |
|---|---|---|---|---|---|---|---|---|---|---|---|
| SGD | ✓ | (✓) | (✓) | ✓ | | | | | | | |
| Berahas et al. (2021b) Fang et al. (2024) Curtis et al. (2024a) | ✓ | – | ✓ | – | – | ✓ | – | – | – | ✓ | – |
| Na et al. (2023a) | ✓ | – | ✓($\mathcal{C}^3$) | – | – | ✓ | – | – | – | ✓($\mathcal{C}^3$) | – |
| Shi et al. (2022) Curtis et al. (2024b) | ✓ | – | ✓ | – | – | (✓) | ✓ | – | – | ✓ | – |
| Na et al. (2023b) | ✓ | – | ✓($\mathcal{C}^2$) | – | – | (✓) | ✓ | – | – | ✓($\mathcal{C}^2$) | – |
| Bollapragada et al. (2023) | ✓ | – | ✓(+ cvx) | – | – | ✓ | – | ✓ | – | – | – |
| Oztoprak et al. (2023) | ✓ | – | ✓ | – | ✓ | ✓ | – | – | – | ✓ | – |
| SSL-ALM Huang et al. (2025) | ✓ | – | ✓ | – | ✓ | (✓) | ✓ | ✓ | – | – | – |
| Stoch. Ghost Facchinei & Kungurtsev (2023) | ✓ | – | ✓ | – | ✓ | (✓) | ✓ | – | – | ✓ | – |
| Stoch. Switch. Subg. Huang & Lin (2023) | ✓ | ✓ | – | – | ✓ | (✓) | ✓ | – | ✓ | – | – |

distinguish between the random variable $\xi$ associated with the objective function and the random variable $\zeta$ associated with the constraint function. Their probability distributions are denoted by $\mathcal{P}_\xi$ and $\mathcal{P}_\zeta$. For an integer $J \in \mathbb{N}$, a set $\{\xi_j\}_{j=1}^J$ of independent and identically distributed random variables $\xi_1, \ldots, \xi_J \overset{iid}{\sim} \mathcal{P}_\xi$ is called a mini-batch. Inspired by Na et al. (2023a), we use the following notation for the stochastic estimates computed from a mini-batch of size $J$:

$$\overline{\nabla}^J f(x) = \tfrac{1}{J} \sum_{j=1}^J \nabla f(x, \xi_j), \quad \bar{c}^J(x) = \tfrac{1}{J} \sum_{j=1}^J c(x, \zeta_j), \quad \overline{\nabla}^J c(x) = \tfrac{1}{J} \sum_{j=1}^J \nabla c(x, \zeta_j). \quad (6)$$

### 3.1 REVIEW OF METHODS FOR CONSTRAINED ERM

We compare recent constrained optimization algorithms considering a stochastic objective function in Table 3. We note that most of them do not consider the case of stochastic constraints. Among those which do consider stochastic constraints, only three admit inequality constraints. Moreover, with the exception of Huang & Lin (2023), all the algorithms in Table 3 assume $F$ to be at least $\mathcal{C}^1$, which makes addressing the challenge of nonsmoothness of $F$ infeasible. Davis et al. (2020) leads us to conclude that assuming the objective and constraint functions to be tame and locally Lipschitz is a suitable requirement for solving (5) with theoretical guarantees of convergence. At this point, however, no such algorithm exists, to the best of our knowledge.

Consequently, we consider the practical performance of the algorithms that address the challenges of solving (5) most closely: Stochastic Ghost, SSL-ALM, and Stochastic Switching Subgradient.

### 3.2 STOCHASTIC GHOST METHOD (STGH)

Facchinei & Kungurtsev (2023) propose the Stochastic Ghost method, that combines a deterministic method for solving (1) (Facchinei et al., 2021) with a stochastic sampling approch for nonlinear maps (Blanchet et al., 2019). The deterministic method of Facchinei et al. (2021) consists in solving subproblem (7) to obtain a direction $d$, and then to preform a line search. Here, $e \in \mathbb{R}^m$ is a vector with all elements equal to one, $\tau$ and $\beta > 0$ are user-prescribed constants, and $\kappa_k$ is defined as a certain convex combination of optimization subproblems related to $C$ and $\nabla C$. The definition of $\kappa_k$ enables to expand the feasibility region so that (7) is always feasible. As the problem (1) is stochastic, the subproblem (7) is modified to a stochastic version (8), using the notation in (6):

$$\begin{aligned} \min_d \quad & \nabla F(x_k)^\top d + \tfrac{\tau}{2}\|d\|^2, \\ \text{s.t.} \quad & C(x_k) + \nabla C(x_k)^\top d \leq \kappa_k e, \\ & \|d\|_\infty \leq \beta, \end{aligned} \quad (7)$$

$$\begin{aligned} \min_d \quad & \overline{\nabla}^J f(x_k)^\top d + \tfrac{\tau}{2}\|d\|^2, \\ \text{s.t.} \quad & \bar{c}^J(x_k) + \overline{\nabla}^J c(x_k)^\top d \leq \overline{\kappa_k}^J e, \\ & \|d\|_\infty \leq \beta. \end{aligned} \quad (8)$$

In the stochastic setting (8), an unbiased estimate $d(x_k)$ of the line search direction $d$ is computed using four particular mini-batches as follows. To facilitate comprehension, we denote $X_k^J = \{X_{k,j}\}_{j=1}^J$ a mini-batch of size $J$ with the $j$-th element $X_{k,j} = (\nabla f(x_k, \xi_{k,j}), c(x_k, \zeta_{k,j}), \nabla c(x_k, \zeta_{k,j}))$. First, we sample a random variable $N \sim \mathcal{G}(p_0)$ from the geometric distribution. Then we sample the mini-batches $X_k^1$ and $X_k^{2^{N+1}}$ and we partition the mini-batch $X_k^{2^{N+1}}$ of size $2^{N+1}$ into two mini-batches $\text{odd}(X_k^{2^{N+1}})$ and $\text{even}(X_k^{2^{N+1}})$ of size $2^N$. Finally, we solve (8) for each of the four mini-batches, denoting by $d(x_k; X_k^J)$ the solution of (8) for the corresponding mini-batch $X_k^J$. We obtain

$$d(x_k) = \frac{d(x_k; X_k^{2^{N+1}}) - \frac{1}{2}\left(d(x_k; \text{odd}(X_k^{2^{N+1}})) + d(x_k; \text{even}(X_k^{2^{N+1}}))\right)}{(1-p_0)^N p_0} + d(x_k; X_k^1). \quad (9)$$

An update between the iterations $x_k$ and $x_{k+1}$ is then computed as $x_{k+1} = x_k + \alpha_k d(x_k)$, where the deterministic stepsize $\alpha_k$ should be square-summable $\sum_{k=1}^\infty \alpha_k^2 < \infty$ but not summable $\sum_{k=1}^\infty \alpha_k = \infty$. For more details, see Algorithm 1 (Appendix C).

### 3.3 STOCHASTIC SMOOTHED AND LINEARIZED AL METHOD (SSL-ALM)

The Stochastic Smoothed and Linearized AL Method (SSL-ALM) was described in Huang et al. (2025) for optimization problems with stochastic linear constraints. Although problem (1) has non-linear inequality constraints, we use the SSL-ALM due to the lack of algorithms in the literature dealing with stochastic non-linear constraints; see Table 3. The transition between equality and inequality constraints is handled with slack variables. Following the structure of Huang et al. (2025), we minimize over the set $\mathcal{X} = \mathbb{R}^n \times \mathbb{R}_{\geq 0}^m$. The method is based on the augmented Lagrangian (AL) function $L_\rho(x, y) = F(x) + y^\top C(x) + \frac{\rho}{2}\|C(x)\|^2$; see e.g., (Bertsekas & Rheinboldt, 2014). Adding a smoothing term with an additional variable $z \in \mathbb{R}^n$ yields the proximal AL function

$$K_{\rho,\mu}(x, y, z) = L_\rho(x, y) + \frac{\mu}{2}\|x - z\|^2.$$

The SSL-ALM method was originally proposed in Huang et al. (2025) where it is interpreted as an inexact gradient descent step on the Moreau envelope. An important property of the Moreau envelope is that its stationary points coincide with those of the original function.

The strength of this method is that, as opposed to the Stochastic Ghost method, it does not use large mini-batch sizes. In each iteration, we sample $\xi \overset{iid}{\sim} \mathcal{P}_\xi$ to evaluate the objective and $\zeta_1, \zeta_2 \overset{iid}{\sim} \mathcal{P}_\zeta$ to evaluate the constraint function and its Jacobian matrix, respectively. The function

$$G(x, y, z; \xi, \zeta_1, \zeta_2) = \nabla f(x, \xi) + \nabla c(x, \zeta_1)^\top y + \rho \nabla c(x, \zeta_1)^\top c(x, \zeta_2) + \mu(x - z) \quad (10)$$

is defined so that, in iteration $k$, $\mathbb{E}_{\xi, \zeta_1, \zeta_2}[G(x_k, y_{k+1}, z_k; \xi, \zeta_1, \zeta_2)] = \nabla K_{\rho,\mu}(x_k, y_{k+1}, z_k)$. Denoting $\eta, \tau$, and $\beta$ positive parameters, the update is

$$\begin{aligned}
y_{k+1} &= y_k + \eta c(x, \zeta_1), \\
x_{k+1} &= \text{proj}_\mathcal{X}(x_k - \tau G(x_k, y_{k+1}, z_k; \xi, \zeta_1, \zeta_2)), \\
z_{k+1} &= z_k + \beta(x_k - z_k).
\end{aligned} \quad (11)$$

For more details, see Algorithm 2 (Appendix C).

### 3.4 STOCHASTIC SWITCHING SUBGRADIENT METHOD (SSW)

The Stochastic Switching Subgradient method was described in Huang & Lin (2023) for optimization over a closed convex set $\mathcal{X} \subset \mathbb{R}^d$ which is easy to project on. It allows for weakly-convex, possibly nonsmooth, objective and constraint functions. They consider subgradients instead of gradients.

The algorithm relies on a prescribed sequence of infeasibility tolerances $\epsilon_k$ and of stepsizes $\eta_k^f$ and $\eta_k^c$. At iteration $k$, we sample $\zeta_1, \ldots, \zeta_J \overset{iid}{\sim} \mathcal{P}_\zeta$ to compute $\bar{c}^J(x_k)$. If $\bar{c}^J(x_k)$ is smaller than $\epsilon_k$, we sample $\xi \overset{iid}{\sim} \mathcal{P}_\xi$ and update using a stochastic estimate $S^f(x_k, \xi)$ of a subgradient of $F$:

$$x_{k+1} = \text{proj}_\mathcal{X}(x_k - \eta_k^f S^f(x_k, \xi)).$$

If not, we sample $\zeta \overset{iid}{\sim} \mathcal{P}_\zeta$ and update using a stochastic estimate $S^c(x_k, \zeta)$ of a subgradient of $C$:

$$x_{k+1} = \text{proj}_{\mathcal{X}}(x_k - \eta_k^c S^c(x_k, \zeta)).$$

In either case, the updates are only saved starting from a prescribed index $k_0$ and the final output is sampled randomly from the saved updates. The algorithm presented here is slightly more general than the one presented in Huang & Lin (2023): we allow for different stepsizes for the objective and the constraint update, while the original method employs equal stepsizes $\eta_k^f = \eta_k^c$. For more details, see Algorithm 3 (Appendix C).

## 4  EXPERIMENTAL EVALUATION

In this section, we illustrate the presented algorithms on a real-world instance of the ACS dataset, comparing how they fare with optimization and fairness metrics.

### 4.1  DATASET FOR FAIR ML

Ding et al. (2021) proposed a large-scale dataset for fair Machine Learning, based on the ACS PUMS data sample (American Community Survey Public Use Microdata Sample). The ACS survey is sent annually to approximately 3.5 million US households in order to gather information on features such as ancestry, citizenship, education, employment, or income. Therefore, it has the potential to give rise to large-scale learning and optimization problems.

In our experiments, we use the ACSIncome dataset, and choose the binary classification task of predicting whether an individual's income is over $50,000.

### 4.2  EXPERIMENTS

**Numerical setup.**  Experiments are conducted on an Asus Zenbook UX535 laptop with AMD Ryzen 7 5800H CPU, and 16GB RAM, using Python with the PyTorch package (Paszke et al., 2019).

#### 4.2.1  BINARY PROTECTED ATTRIBUTE

**Dataset and problems**  We use the ACSIncome dataset over the state of Oklahoma.The dataset contains 9 features and 17,917 data points, and may be accessed via the Python package Folktables. We choose race (**RAC1P**) as the protected attribute. In the original dataset, it is a categorical variable with 9 values. For the purposes of this experiment, we binarize it to obtain the non-protected group of "white" people and the protected group of "non-white" people. The dataset is split randomly into train (80%, 14,333 points) validation (10%, 1,792), and test (10%, 1.792 points) subsets and it is stratified with respect to the protected attribute, i.e., the proportion of "white" and "non-white" samples in the training, validation, and test sets is equivalent to that in the full dataset (30.8% of positive labels in group "white", 20.7% in the group "non-white"). The protected attributes are then removed from the data so that the model cannot learn from them directly. The data is normalized using Scikit-Learn StandardScaler.

Note that ACSIncome is a real-world dataset for which ERM-based predictors without fairness safeguards are known to learn biases (Han et al., 2023). Accordingly, Table 4 (line 1) shows that an ERM predictor without fairness safeguards has poor fairness metrics; see also Figure 4.

**Problems.**  We consider the constrained ERM problem (4) without any regularization $\mathcal{R} = 0$. As our data is divided into just two groups, we constrain the difference between the loss values $\ell^{s_i}$ directly, instead of taking the average. In addition, we consider as baselines the ERM problem (2) without any regularization, $\mathcal{R} = 0$, and with a fairness inducing regularizer $\mathcal{R}$ that promotes small difference in accuracy between groups, provided by the Fairret library (Buyl et al., 2024). In all problems, we take as loss function the Binary Cross Entropy with Logits Loss

$$\ell(f_\theta(X_i), Y_i) = -Y_i \cdot \log \sigma(f_\theta(X_i)) - (1 - Y_i) \cdot \log(1 - \sigma(f_\theta(X_i))), \tag{12}$$

where $\sigma(z) = \frac{1}{1+e^{-z}}$ is the sigmoid function, and the prediction function $f_\theta$ is a neural network with 2 interconnected hidden layers of sizes 64 and 32 and ReLU activation, with a total of 194 parameters.

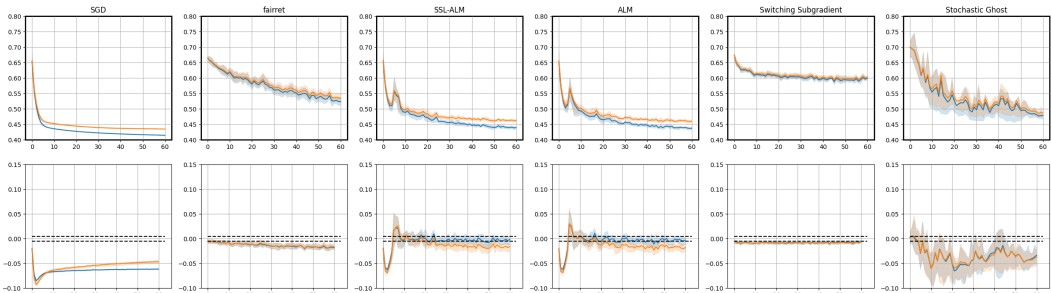

Figure 1: Train (blue) and test (orange) statistics over time (s) on the ACS Income dataset for each algorithm: SGD (column 1), fairret-regularized SGD (column 2), SSL-ALM (column 3), ALM (column 4) Switching Subgradient (column 5), and Stochastic Ghost (column 6). The plots depict the mean values for loss (first row) and the constraint at each timestamp, rounded to the nearest 0.5 seconds, over 10 runs. The shaded area depicts the region between the first and third quartiles.

**Algorithms and parameters.** We assess the performance of four algorithms for solving the constrained problem (4):

- Stochastic Ghost (StGh) (Sec. 3.2 - parameters $\alpha_0 = 0.05$, $\rho = 0.8$, $\tau = 1$, $\beta = 20$, $\lambda = 0.5$, $\hat{\alpha} = 0.05$)
- SSL-ALM (Sec. 3.3 - parameters $\mu = 2.0$, $\rho = 1.0$, $\tau = 0.01$, $\eta = 0.05$, $\beta = 0.5$, $M_y = 10$),
- plain Augmented Lagrangian Method ALM (Sec. 3.3, smoothing term removed $\mu = 0$, otherwise the same setting as SSL-ALM),
- Stochastic Switching Subgradient (SSw) (Sec. 3.4 - $\eta^f = 0.05$, $\eta^c = 0.04$, $\epsilon_0 = 0.01$, $\epsilon_k = \frac{\epsilon_0}{\sqrt{k+1}}$).

The hyperparameter values were chosen by grid-search on a validation set; we report the performance of the algorithms under different hyperparameter choices in appendix B.

We also provide the behavior of SGD for solving the ERM problem, both with no fairness safeguards (SGD), and with fairness regularization on accuracy as provided by the Fairret library (Buyl et al., 2024) (SGD-Fairret). These methods serve as baselines. When estimating the constraints, we sample an equal number of data points for every subgroup.

**Optimization performance.** Figure 1 present the evolution of loss and constraint values over the train and test datasets for the four algorithms addressing the constrained problem (columns 3–6), as well as for the two baselines: SGD without fairness (col. 1), and SGD with fairness regularization (col. 2). Each algorithm is run 10 times, and the plots display the mean and quartiles values.

To a certain extent, the four algorithms (col. 3–6) succeed in minimizing the loss and satisfying the constraints on the train set. The AL-based methods (col. 3 and 4) demonstrate a better behavior compared to StGh (col. 5) and SSw (col. 6). Indeed, StGh exhibits higher variability in both loss and constraint values. SSw satisfies the constraint the best out of all methods, but fails to minimize the objective function to the extent that other algorithms do. See appendix B for exploration of the algorithms' behaviour under different hyperparameter choices.

SGD (col. 1) exhibits the lowest variability in the trajectory and minimizes the loss in least time, but, as expected, does not satisfy the constraints. Fairret-regularized SGD (col. 2) minimizes the objective function slower than some constrained algorithms, while maintaining a small, but increasing, constraint violation. Note that the penalty parameter was optimized for constraint satisfaction; other penalty values would allow a faster minimization of the objective at the cost of higher constraint violation; see appendix B more for details. This observation is consistent with Ramirez et al. (2025).

The ALM and SSL-ALM schemes satisfy the constraints on the train set. On the test set, however, they are slightly biased towards negative values. Such bias is expected on unseen data and reflects the generalization behavior of fairness-constrained estimators. This is beyond the scope of the current work; see e.g. Chamon et al. (2022).

**Fairness performance.** Table 4 displays the fairness metrics presented in Section 2: independence (Ind), separation (Sp), and sufficiency (Sf), along with inaccuracy (Ina). The mean value and standard

Table 4: Fairness metrics (independence, separation, sufficiency), inaccuracy, and Wasserstein distances between groups (Wd) for the four constrained estimators and the two baselines.

| | Train | | | | | Test | | | | |
| Algname | Ind | Sp | Ina | Sf | Wd | Ind | Sp | Ina | Sf | Wd |
|---|---|---|---|---|---|---|---|---|---|---|
| SGD | $0{,}095_{\pm0,003}$ | $0{,}124_{\pm0,006}$ | $\mathbf{0{,}186_{\pm0,023}}$ | $0{,}065_{\pm0,004}$ | $0{,}062_{\pm0,012}$ | $0{,}098_{\pm0,005}$ | $0{,}155_{\pm0,015}$ | $\mathbf{0{,}209_{\pm0,019}}$ | $0{,}061_{\pm0,005}$ | $0{,}183_{\pm0,015}$ |
| StGh | $0{,}082_{\pm0,020}$ | $0{,}103_{\pm0,052}$ | $0{,}230_{\pm0,025}$ | $\mathbf{0{,}048_{\pm0,022}}$ | $0{,}037_{\pm0,038}$ | $0{,}086_{\pm0,024}$ | $0{,}123_{\pm0,059}$ | $0{,}239_{\pm0,020}$ | $0{,}057_{\pm0,021}$ | $0{,}161_{\pm0,053}$ |
| ALM | $0{,}083_{\pm0,009}$ | $0{,}112_{\pm0,024}$ | $0{,}210_{\pm0,010}$ | $0{,}057_{\pm0,011}$ | $0{,}063_{\pm0,021}$ | $\mathbf{0{,}058_{\pm0,012}}$ | $0{,}114_{\pm0,014}$ | $0{,}244_{\pm0,007}$ | $0{,}221_{\pm0,017}$ | $0{,}158_{\pm0,027}$ |
| SSL-ALM | $\mathbf{0{,}074_{\pm0,005}}$ | $\mathbf{0{,}091_{\pm0,010}}$ | $0{,}208_{\pm0,009}$ | $0{,}050_{\pm0,009}$ | $0{,}054_{\pm0,014}$ | $0{,}083_{\pm0,006}$ | $\mathbf{0{,}108_{\pm0,018}}$ | $0{,}223_{\pm0,013}$ | $\mathbf{0{,}046_{\pm0,012}}$ | $0{,}170_{\pm0,033}$ |
| SSw | $0{,}096_{\pm0,010}$ | $0{,}139_{\pm0,011}$ | $0{,}191_{\pm0,020}$ | $0{,}064_{\pm0,007}$ | $\mathbf{0{,}001_{\pm0,001}}$ | $0{,}103_{\pm0,018}$ | $0{,}168_{\pm0,036}$ | $0{,}212_{\pm0,020}$ | $0{,}066_{\pm0,019}$ | $\mathbf{0{,}018_{\pm0,005}}$ |
| SGD-Fairret | $0{,}091_{\pm0,011}$ | $0{,}128_{\pm0,016}$ | $0{,}190_{\pm0,020}$ | $0{,}059_{\pm0,010}$ | $0{,}004_{\pm0,003}$ | $0{,}091_{\pm0,016}$ | $0{,}141_{\pm0,028}$ | $0{,}211_{\pm0,019}$ | $0{,}056_{\pm0,014}$ | $0{,}059_{\pm0,021}$ |

deviation over 10 runs are presented for the four fairness-constrained models and the two baselines, both on train and test sets. For all metrics, smaller is better. We provide additional details in appendices A and B.1.

Among the four fairness-constrained models, StGh performs best in terms of sufficiency, but worst in terms of accuracy. Overall, the constrained optimization models improve on the fairness metrics of the unconstrained SGD and the regularized SGD-Fairret models. SSw has both fairness metrics and inaccurary comparable to that of the unconstrained SGD model. This is consistent with the observation that the optimization method, with our choice of parameters, favored minimizing the objective over satisfying the constraints. The ALM and SSL-ALM methods provide the best compromise: they improve independence, separation, and sufficiency relative to the SGD model, while moderately degrading accuracy. SGD-Fairret slightly improves sufficiency relative to the SGD model. Similar observations hold for metrics on the test set.

### 4.2.2 MULTI-VALUED PROTECTED ATTRIBUTE

**Dataset and problems.** We again consider the dataset ACSIncome, but over the state of Virginia, and choose `Mariage` as the protected attribute. This attribute takes fives values, as opposed to the binary attribute setup of section 4.2.1.

We consider three optimization problems as approaches to tackle the learning task. First, we consider the *constrained* learning problem as described in eq. (4), with $m = 5$. Second, we consider the unconstrained, but *penalized*, problem

$$\min_{\theta\in\mathbb{R}^n} \frac{1}{N}\sum_{i=1}^{N}\ell(f_\theta(X_i),Y_i) + \mathcal{R}(\theta) + \lambda\sum_{i=1}^{m}\left|\ell^{s_i}(\theta) - \frac{1}{m}\sum_{j=1}^{m}\ell^{s_j}(\theta)\right|, \tag{13}$$

where $\lambda$ is a penalization weight. Third, we consider the unconstrained and unpenalized problem, as described in eq. (2), for comparison.

**Algorithms and parameters.** We solve the constrained learning problem (4) with Stochastic Ghost, Switching Subgradient, and SSL-ALM. We solve the baseline penalized problem (13) and the basic unconstrained unpenalized problem (2) using SGD.

The hyperparameters for each algorithm were tuned on the validation set; we picked the values resulting in lowest loss and constraint satisfaction after 60 seconds. Our hyperparameter choices are:

- Regularized SGD: $\lambda = 0.4$.
- SSL-ALM: $\tau = \eta = 0.01, \beta = 0.5, \mu = 2, \rho = 1$.
- Stochastic Ghost: $\beta = 1.0, \gamma_0 = 0.005, \zeta = 0.05, \rho = 0.1, \tau = 1, \lambda = 0.5$.
- SSw: $\eta^f = 0.05$ constant, $\eta_k^c$ diminishing with $\eta_0^c = 0.25, \eta_k^c = \frac{\eta_{k-1}^c}{\sqrt{k}}, k > 0$; constraint tolerance $\epsilon$ diminishing with $\epsilon_0 = 0.01, \epsilon_k = \frac{\epsilon_{k-1}}{\sqrt{k}}, k > 0$.

**Optimization performance.** We report in Figure 2 the evolution of the mean and quartiles of the train and test values over 10 runs. SGD on the unconstrained and unpenalized problem (2) (first row) converges to a model such that the constraint are consistently above the constraint bound for three values of the protected attribute (Wid, Div, and Nev). SGD on the penalized problem (second row) manages to meet all constraints for the training set, but constraint Div on the test set is eventually violated. The three constrained methods minimize function values while keeping with the constraint

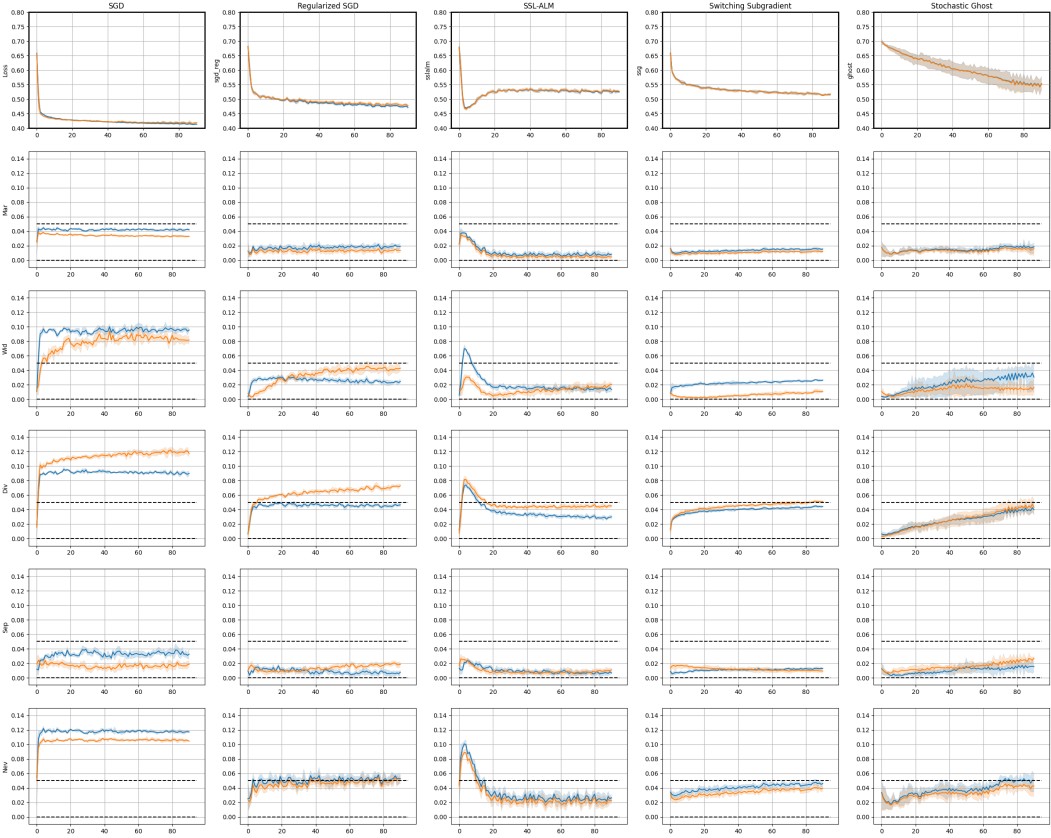

Figure 2: Train (blue) and test (orange) statistics over time (s) on the ACS Income dataset for each algorithm: SGD (column 1), regularized SGD (column 2), SSL-ALM (column 3), Switching Subgradient (column 4), and Stochastic Ghost (column 5). The plots depict the mean values for loss (first row) and constraints (second to last row) at each timestamp, rounded to the nearest 0.5 seconds, over 10 runs. The shaded area depicts the region between the first and third quartiles.

bounds. The performance of SGD on penalized problem and the three constrained algorithms is comparable.

Note that the penalized problem required consequent preliminary computations in order to tune the penalization parameter $\lambda$. We found that the performance of the estimator was sensitive to the value of $\lambda$ in two aspects: (i) on the difficulty of the optimization problem for SGD, and (ii) in the trade-off between minimization of the empirical risk and satisfaction of constraints; see appendix B.2 for details. In contrast, the algorithms for constrained minimization showed (i) a smaller sensitivity to their hyperparameters, particularly so for SSL-ALM, and (ii) a controllable trade-off between minimization of the empirical risk, and constraint satisfaction; again, see appendix B.2 for details. This observation is consistent with the argument of Ramirez et al. (2025).

## 5 CONCLUSION

To the best of our knowledge, this paper provides the first benchmark for assessing the performance of optimization methods on real-world instances of fairness constrained training of models. We highlight the challenges of this approach, namely that objective and constraints are non-convex, non-smooth, and large-scale, and review the performance of four practical algorithms.

### LIMITATIONS

Our work identifies that there is currently no algorithm with guarantees for solving the fairness constrained problem. Above all, we hope that this work, along with the Python toolbox for easy

benchmarking of new optimization methods, will stimulate further interest in this topic. Also, we caution readers that the method present here is not a silver-bullet that handles all biases and ethical issues of training ML models. In particular, care must be taken that fair ML is part of a interdisciplinary pipeline that integrates the specifics of the use-case, and that it does not serve as an excuse for pursuing Business-As-Usual policies that fail to tackle ethical issues (Balayn et al., 2023; Wachter et al., 2021).

## REPRODUCIBILITY

Code to reproduce the experiments is provided in the Supplementary Material. This includes a readme file with instructions to reproduce experiments. Details on the computing environment are provided in Section 4.

## ACKNOWLEDGMENTS

The authors acknowledge the support of National Recovery Plan funded project MPO 60273/24/21300/21000 CEDMO 2.0 NPO. This work was supported in part by the European Unions Horizon Europe research and innovation programme under grant agreements 101084642 (Codiet) and 101070568 (AutoFair).

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

## A   ADDITIONAL DETAILS ON THE FAIRNESS OF THE BINARY EXPERIMENT.

Figure 3 presents the distribution of predictions over both groups. The distribution of prediction without fairness guarantees (col. 1) clearly does not meet the group fairness standard. Indeed, the "non-white" group has a significantly higher likelihood than the "white" group of receiving small predicted values, and the converse holds for large predicted values. Among the fairness-constrained models, the ALM and SSL-ALM distributions are the closest to the distributions of SGD without fairness, which is consistent with retaining good prediction information. The Fairret penalized formulation (col. 2) and SSw (col. 5) have a center-heavy distribution, which, in this case, is evidence of poor objective minimization.

The SSL-ALM, ALM, and Stochastic Ghost (col. 3, 4, and 6) have closer distributions across groups than the unconstrained formulation.

Numerically, this is expressed in Table 4 (col. Wd), which reports the value of the Wasserstein distance between group distributions for each model.

Table 4 displays the fairness metrics presented in Section 2: independence (Ind), separation (Sp), and sufficiency (Sf), along with inaccuracy (Ina). The mean value and standard deviation over 10 runs are presented for the four fairness-constrained models and the two baselines, both on train and test sets. Figure 4 presents the mean values as spider plots. For all metrics, smaller is better.

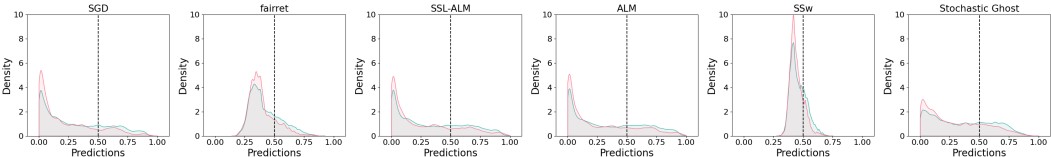

Figure 3: Distribution of predictions for each algorithm. Left to right: SGD, SGD-Fairret, SSL-ALM, ALM, SSw, StGh, Blue and red denote "white" and "non-white" groups.

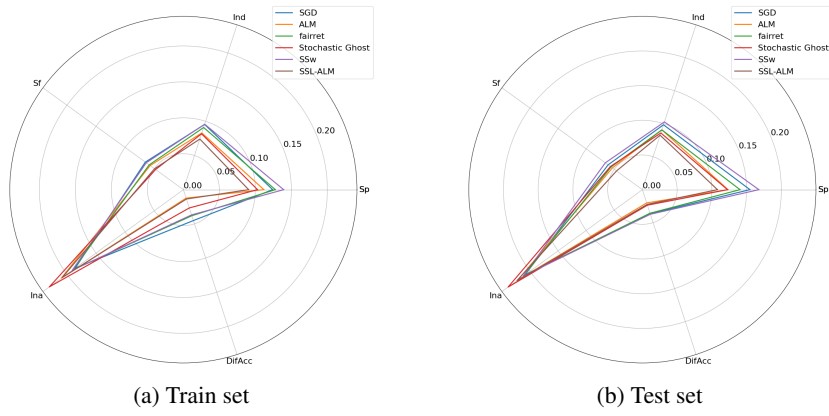

(a) Train set                                      (b) Test set

Figure 4: Average value of the three fairness metrics (independence (Ind), separation (Sp), and sufficiency (Sf)), along with mean inaccuracy (Ina), and difference in accuracy between the two groups (DifAcc). For all metrics, smaller values are better.

## B   HYPERPARAMETER SENSITIVITY ANALYSIS

In this section, we present the performance of algorithms on the validation set, justifying our hyperparameter choices in the experimental section. For each algorithm, we provide a table of loss and constraint violation values under different hyperparameter choices after a training run.

For each algorithm, we perform 5 runs with each hyperparameter combination. Every 100 iterations, we save the model state (except for Stochastic Ghost, where we save every 10 iterations due to higher iteration cost). We then pick our hyperparameters based on the average loss value and constraint

violation over the last 20 states over 5 runs. For algorithms that exhibit exceptionally noisy trajectories, we also provide convergence plots.

## B.1 EXPERIMENT 1 - BINARY ATTRIBUTE

We recall that the constraint bound $\delta$ was set to $0.005$. With random parameter initialization, initial loss value was $\approx 0.7$.

### B.1.1 STOCHASTIC SWITCHING SUBGRADIENT

Stochastic Switching Subgradient admits a choice of constant, diminishing, or adaptive stepsizes for both objective and constraint updates ($\eta^f$ and $\eta^c$, respectively). In this experiment, we were unable to find a combination of the above that would enable the algorithm to both minimize the objective on par with other methods and maintain the feasibility of the solution. The run results are presented in table 5.

As suggested in the original paper, the diminishing stepsize follows the rule

$$\eta_k = \frac{\eta_0}{\sqrt{k+1}}$$

and the adaptive stepsize follows the rule

$$\eta_k = \frac{c(x_k, \zeta)}{||S^c(x_k, \zeta)||^2}$$

We use diminishing infeasibility tolerance, with $\epsilon_0 = 0.01, \epsilon_k = \frac{\epsilon_0}{\sqrt{k+1}}$

| Hyperparameters | Loss | | Constraint violation | |
|---|---|---|---|---|
| | mean | std | mean | std |
| $\eta^f$ adaptive, $\eta^c$ adaptive | 0.681 | 0.006 | 0.000 | 0.000 |
| $\eta^f = 0.05, \eta^c$ adaptive | 0.679 | 0.006 | 0.000 | 0.000 |
| $\eta^f = 0.05, \eta^c = 0.05$ | 0.626 | 0.006 | 0.000 | 0.000 |
| $\eta^f = 0.05, \eta^c = 0.045$ | 0.589 | 0.01 | 0.0012 | 0.002 |
| $\eta^f = 0.05, \eta^c = 0.04$ | 0.567 | 0.014 | 0.003 | 0.003 |
| $\eta^f = 0.05, \eta^c = 0.025$ | 0.469 | 0.012 | 0.018 | 0.004 |
| $\eta^f = 0.05, \eta^c$ dimin with $\eta_0^c = 0.05$ | 0.409 | 0.001 | 0.025 | 0.004 |
| $\eta^f = 0.05, \eta^c$ dimin with $\eta_0^c = 0.25$ | 0.409 | 0.001 | 0.025 | 0.004 |
| $\eta^f = 0.05, \eta^c$ dimin with $\eta_0^c = 0.5$ | 0.410 | 0.002 | 0.027 | 0.005 |
| $\eta^f = 0.05, \eta^c$ dimin with $\eta_0^c = 3.0$ | 0.476 | 0.016 | 0.018 | 0.004 |
| $\eta^f$ dimin with $\eta_0^f = 0.05, \eta^c$ dimin with $\eta_0^c = 0.05$ | 0.660 | 0.002 | 0.000 | 0.000 |
| $\eta^f$ dimin with $\eta_0^f = 0.5, \eta^c$ dimin with $\eta_0^c = 0.5$ | 0.633 | 0.004 | 0.000 | 0.000 |

Table 5: Loss and constraint violation on the validation set after 5 30-second runs of the Stochastic Switching Subgradient in the setup of Exp. 1, rounded to 3 digits.

We pick $\eta^f = 0.05, \eta^c = 0.04$, as it results in the best objective value with relatively low infeasibility; we then conduct

### B.1.2 SSL-ALM

SSL-ALM allows the user to tune the primal ($\tau$) and dual ($\eta$) parameter update rate, the penalty multiplier $\rho$, as well as the smoothing update rate $\beta$ and multiplier $\mu$. We consider only constant stepsizes $\tau$ and $\eta$, and fix $\rho = 1$. The run results are presented in table 6.

The lowest loss value with the least constraint violation is offered by $\tau = 0.01, \eta = 0.05, \mu = 0$; the second best combination is $\tau = 0.01, \eta = 0.05, \mu = 2$. As the original paper assumes $\mu \geq 2$, we test both options as "ALM" and "SSL-ALM", respectively.

| Hyperparameters | | | Loss | | Constraint violation | |
|---|---|---|---|---|---|---|
| $\tau$ | $\eta$ | $\mu$ | mean | std | mean | std |
| 0.05 | 0.05 | 2 | 0.4376 | 0.0283 | 0.0194 | 0.0294 |
| 0.05 | 0.01 | 2 | 0.4240 | 0.0096 | 0.0112 | 0.0097 |
| 0.01 | 0.05 | 2 | 0.4465 | 0.0122 | 0.0074 | 0.0085 |
| 0.01 | 0.01 | 2 | 0.4416 | 0.0138 | 0.0133 | 0.0145 |
| 0.05 | 0.05 | 0 | 0.4560 | 0.0962 | 0.0282 | 0.0620 |
| 0.05 | 0.01 | 0 | 0.4222 | 0.0099 | 0.0111 | 0.0100 |
| 0.01 | 0.05 | 0 | 0.4448 | 0.0114 | 0.0070 | 0.0078 |
| 0.01 | 0.01 | 0 | 0.4407 | 0.0155 | 0.0157 | 0.0154 |
| 0.05 | 0.05 | 4 | 0.4377 | 0.0242 | 0.0180 | 0.0271 |
| 0.05 | 0.01 | 4 | 0.4252 | 0.0098 | 0.0121 | 0.0095 |
| 0.01 | 0.05 | 4 | 0.4471 | 0.0112 | 0.0073 | 0.0077 |
| 0.01 | 0.01 | 4 | 0.4387 | 0.0132 | 0.0143 | 0.0099 |

Table 6: Loss and constraint violation on the validation set after 5 30-second runs of the SSL-ALM variants in the setup of Exp. 1, rounded to 4 digits.

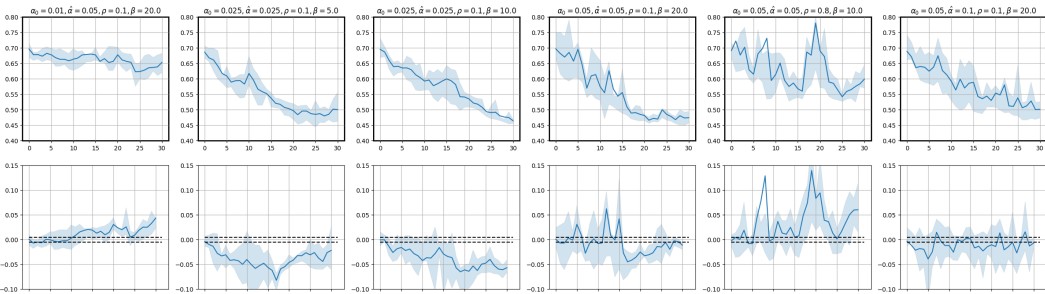

Figure 5: Loss (top row) and constraint (bottom row) evolution of the Stochastic Ghost on the validation dataset with different hyperparameter choices. The line corresponds to the mean value over 5 runs, the shaded region - to the area between the 1st and 3rd quartiles over 5 runs.

### B.1.3    STOCHASTIC GHOST

Among the algorithms discussed, Stochastic Ghost features the largest number of hyperparameters, which makes the algorithm difficult to tune. We set $p_0$, the parameter of the geometric distribution that controls the number of samples taken at each iteration, to $0.4$, as, in expectation, this matches the batch size used for the other algorithms. As in the original paper Facchinei & Kungurtsev (2023), we use the OSQP solver to solve the subproblems.

We attempt to tune the stepsize $\alpha_0$, the stepsize decay rate $\hat{\alpha}$, and two subproblem parameters $\beta$ and $\rho$.

The run results are presented in table 7. Due to the level of noise in the algorithm's trajectories, we add the plots for some of the more stable hyperparameter configurations in fig. 5.

We pick the values $\alpha_0 = 0.05, \hat{\alpha} = 0.05, \rho = 0.1, \beta = 20$.

### B.1.4    SGD+FAIRRET

For regularized SGD, we tune the regularization penalty multiplier; once we find a value that leads to minimizing both loss and constraint violation, we try different stepsize values.

The run results are presented in table 8.

We pick a multiplier equal to 3 and LR of 0.05.

| Hyperparameters | | | | Loss | | Constraint violation | |
|---|---|---|---|---|---|---|---|
| $\alpha_0$ | $\hat{\alpha}$ | $\rho$ | $\beta$ | mean | std | mean | std |
| 0.0100 | 0.0500 | 0.1000 | 10 | 0.6144 | 0.0287 | **0.0135** | 0.0115 |
| 0.0100 | 0.0500 | 0.1000 | 20 | 0.6390 | 0.0461 | 0.0287 | 0.0297 |
| 0.0100 | 0.0500 | 0.1000 | 5 | 0.6007 | 0.0448 | 0.0256 | 0.0170 |
| 0.0250 | 0.0250 | 0.1000 | 10 | **0.4781** | 0.0245 | 0.0504 | 0.0183 |
| 0.0250 | 0.0250 | 0.1000 | 20 | **0.4674** | 0.0169 | 0.0381 | 0.0175 |
| 0.0250 | 0.0250 | 0.1000 | 5 | 0.4896 | 0.0371 | 0.0367 | 0.0238 |
| 0.0250 | 0.0500 | 0.1000 | 10 | 0.5041 | 0.0402 | 0.0318 | 0.0249 |
| 0.0250 | 0.0500 | 0.1000 | 20 | 0.5246 | 0.0566 | 0.0326 | 0.0236 |
| 0.0250 | 0.0500 | 0.1000 | 5 | 0.5218 | 0.0301 | 0.0343 | 0.0229 |
| 0.0500 | 0.0500 | 0.1000 | 10 | 0.5054 | 0.0295 | 0.0280 | 0.0196 |
| 0.0500 | 0.0500 | 0.1000 | 1 | 0.5020 | 0.0804 | 0.0545 | 0.0510 |
| 0.0500 | 0.0500 | 0.1000 | 20 | **0.4745** | 0.0266 | **0.0152** | 0.0154 |
| 0.0500 | 0.0500 | 0.1000 | 5 | 0.4800 | 0.0338 | 0.0442 | 0.0243 |
| 0.0500 | 0.0500 | 0.8000 | 10 | 0.5700 | 0.0432 | 0.0441 | 0.0375 |
| 0.0500 | 0.0500 | 0.8000 | 20 | 0.6124 | 0.1448 | 0.0851 | 0.0744 |
| 0.0500 | 0.0500 | 0.8000 | 5 | 0.6139 | 0.0783 | 0.0647 | 0.0618 |
| 0.0500 | 0.1000 | 0.1000 | 10 | 0.5136 | 0.0413 | 0.0486 | 0.0297 |
| 0.0500 | 0.1000 | 0.1000 | 20 | 0.5124 | 0.0620 | 0.0379 | 0.0384 |
| 0.0500 | 0.1000 | 0.1000 | 5 | 0.5131 | 0.0505 | 0.0356 | 0.0197 |

Table 7: Loss and constraint violation on the validation set after 5 30-second runs of the Stochastic Ghost in the setup of Exp. 1, rounded to 4 digits.

| Hyperparameters | | Loss | | Constraint violation | |
|---|---|---|---|---|---|
| Multiplier | LR | mean | std | mean | std |
| 0 | 0.05 | 0.406 | 0.003 | 0.020 | 0.004 |
| 1 | 0.05 | 0.406 | 0.004 | 0.010 | 0.007 |
| 2 | 0.05 | 0.421 | 0.008 | 0.012 | 0.006 |
| 3 | 0.05 | 0.539 | 0.026 | 0.004 | 0.005 |
| 3 | 0.07 | 0.523 | 0.029 | 0.006 | 0.008 |
| 3 | 0.1 | 0.523 | 0.033 | 0.007 | 0.008 |
| 4 | 0.05 | 0.676 | 0.009 | 0.001 | 0.002 |

Table 8: Loss and constraint violation on the validation set after 5 30-second runs of fairret-regularized SGD in the setup of Exp. 1, rounded to 3 digits.

## B.2 EXPERIMENT 2 - MULTI-VALUED ATTRIBUTE

We recall that in this experiment, we set the constraint bound to be 0.05. We test Regularized SGD, SSL-ALM, Stochastic Ghost, and Switching Subgradient.

### B.2.1 STOCHASTIC SWITCHING SUBGRADIENT

The run results are presented in table 9. We pick $\eta^f = 0.05$ constant, $\eta_k^c$ diminishing with $\eta_0^c = 0.25$ as it offers the lowest loss value while being close to feasibility.

### B.2.2 SSL-ALM

Unlike Exp. 1, we do not tune the $\mu$ hyperparameter, setting $\mu = 2$. The run results are presented in appendix B.2.2. We pick $\tau = 0.05$, $\eta = 0.01$.

### B.2.3 STOCHASTIC GHOST

The run results are presented in appendix B.2.3. We pick $\alpha_0 = 0.01$, $\hat{\alpha} = 0.2$, $\rho = 0.1$, $\beta = 5$.

| Hyperparameters | | | | Loss | | Max constraint violation | |
|---|---|---|---|---|---|---|---|
| $\eta_0^f$ | $\eta^f$ rule | $\eta_0^c$ | $\eta^c$ rule | mean | std | mean | std |
| 0.05 | const | 0.05 | const | 0.598 | 0.002 | 0.000 | 0.000 |
| 0.5 | const | 0.05 | const | 0.537 | 0.007 | 0.000 | 0.000 |
| 0.05 | const | 0.5 | const | 0.631 | 0.004 | 0.000 | 0.000 |
| 0.5 | const | 0.5 | const | 0.590 | 0.006 | 0.000 | 0.000 |
| 0.05 | const | 0.05 | dimin | 0.543 | 0.018 | 0.031 | 0.001 |
| 0.005 | const | 0.05 | dimin | 0.551 | 0.002 | 0.000 | 0.000 |
| 0.05 | const | 0.05 | dimin | 0.499 | 0.003 | 0.023 | 0.003 |
| 0.05 | const | 0.1 | dimin | 0.506 | 0.006 | 0.011 | 0.003 |
| 0.05 | const | 0.25 | dimin | 0.519 | 0.004 | 0.001 | 0.002 |

Table 9: Loss and constraint violation on the validation set after 5 60-second runs of the Switching Subgradient in the setup of Exp. 2, rounded to 3 digits.

| Hyperparameters | | Loss | | Max constraint violation | |
|---|---|---|---|---|---|
| $\tau$ | $\eta$ | mean | std | mean | std |
| 0.05 | 0.05 | 0.536 | 0.007 | 0.000 | 0.001 |
| 0.05 | 0.01 | 0.530 | 0.006 | 0.000 | 0.001 |
| 0.01 | 0.05 | 0.542 | 0.003 | 0.000 | 0.000 |
| 0.01 | 0.01 | 0.538 | 0.003 | 0.000 | 0.000 |

Table 10: Loss and constraint violation on the validation set after 5 60-second runs of the SSL-ALM in the setup of Exp. 2, rounded to 4 digits.

| Hyperparameters | | | | Loss | | Constraint violation | |
|---|---|---|---|---|---|---|---|
| $\alpha_0$ | $\hat{\alpha}$ | $\rho$ | $\beta$ | mean | std | mean | std |
| 0.0050 | 0.0100 | 0.1000 | 5 | 0.5948 | 0.0218 | 0.0016 | 0.0031 |
| 0.0050 | 0.0100 | 0.1000 | 10 | 0.6149 | 0.0193 | 0.0005 | 0.0012 |
| 0.0050 | 0.0500 | 0.1000 | 5 | 0.6161 | 0.0142 | 0.0001 | 0.0006 |
| 0.0050 | 0.0500 | 0.1000 | 10 | 0.6187 | 0.0191 | 0.0016 | 0.0033 |
| 0.0100 | 0.0100 | 0.1000 | 5 | 0.5373 | 0.0243 | 0.0275 | 0.0296 |
| 0.0100 | 0.0100 | 0.1000 | 10 | 0.5321 | 0.0280 | 0.0290 | 0.0259 |
| 0.0100 | 0.0500 | 0.1000 | 5 | 0.5442 | 0.0222 | 0.0180 | 0.0196 |
| 0.0100 | 0.0500 | 0.1000 | 10 | 0.5583 | 0.0449 | 0.0099 | 0.0166 |
| 0.0100 | 0.0500 | 0.1000 | 20 | 0.5498 | 0.0311 | 0.0106 | 0.0176 |
| 0.0100 | 0.0500 | 0.8000 | 10 | 0.5716 | 0.0235 | 0.0096 | 0.0123 |
| 0.0100 | 0.0500 | 0.8000 | 20 | 0.5571 | 0.0409 | 0.0181 | 0.0184 |
| 0.0100 | 0.1000 | 0.1000 | 5 | 0.5448 | 0.0297 | 0.0197 | 0.0198 |
| 0.0100 | 0.1000 | 0.1000 | 10 | 0.5522 | 0.0213 | 0.0146 | 0.0108 |
| 0.0100 | 0.2000 | 0.1000 | 5 | 0.5868 | 0.0181 | 0.0070 | 0.0071 |
| 0.0100 | 0.2000 | 0.1000 | 10 | 0.5898 | 0.0198 | 0.0016 | 0.0035 |
| 0.0100 | 0.2000 | 0.8000 | 5 | 0.5616 | 0.0330 | 0.0121 | 0.0119 |
| 0.0500 | 0.2000 | 0.1000 | 5 | 0.5144 | 0.0296 | 0.0199 | 0.0255 |

Table 11: Loss and constraint violation on the validation set after 5 60-second runs of the Stochastic Ghost in the setup of Exp. 2, rounded to 4 digits.

### B.2.4 REGULARIZED SGD

The run results are presented in appendix B.2.4. We pick $\lambda = 0.4$.

| Penalty multiplier | Loss | | Constraint violation | |
|---|---|---|---|---|
| | mean | std | mean | std |
| 0 | 0.432 | 0.001 | 0.075 | 0.004 |
| 0.3 | 0.475 | 0.002 | 0.030 | 0.003 |
| 0.4 | 0.512 | 0.002 | 0.001 | 0.002 |
| 0.5 | 0.550 | 0.002 | 0.000 | 0.000 |

Table 12: Loss and constraint violation on the validation set after 5 60-second runs of the Regularized SGD in the setup of Exp. 2, rounded to 4 digits.

## C   ALGORITHMS IN MORE DETAIL

In this section, we provide the pseudocodes of algorithms presented in Section 3 as Algorithms 1 to 3. Recall that we denote by $X_k^J = \{X_{k,j}\}_{j=1}^J$ a mini-batch of size $J$ with the $j$-th element

$$X_{k,j} = (\nabla f(x_k, \xi_{k,j}), c(x_k, \zeta_{k,j}), \nabla c(x_k, \zeta_{k,j})). \tag{14}$$

---

**Algorithm 1** Stochastic Ghost algorithm

---

**Require:** Training dataset $\mathcal{D}$, constraint dataset $\mathcal{C}$, initial neural network weights $x_0$
**Require:** Parameters $p_0 \in (0,1)$, $\alpha_0$, $\hat{\alpha}$, $\rho$, $\tau$, $\beta$
1: **for** Iteration $k = 0$ **to** $K - 1$ **do**
2:    Sample $\xi \overset{iid}{\sim} \mathcal{P}_\xi$ and $\zeta \overset{iid}{\sim} \mathcal{P}_\zeta$
3:    Sample $N \sim \mathcal{G}(p_0)$
4:    Set $J = 2^{N+1}$
5:    Sample a mini-batch $\{\zeta_j\}_{j=1}^J$ so that $\zeta_1, \dots, \zeta_J \overset{iid}{\sim} \mathcal{P}_\zeta$
6:    Sample a mini-batch $\{\xi_j\}_{j=1}^J$ so that $\xi_1, \dots, \xi_J \overset{iid}{\sim} \mathcal{P}_\xi$
7:    Set $X_k^1$ and $X_k^{2^{N+1}}$ using (14)
8:    Compute $d(x_k)$ from (9)
9:    Set $\alpha_k = \alpha_{k-1}(1 - \hat{\alpha}\alpha_{k-1})$
10:    Update $x_{k+1} = x_k + \alpha_k d(x_k)$
11: **end for**

---

**Algorithm 2** Stochastic Smoothed and Linearized AL Method for solving (1)

---

**Require:** Training dataset $\mathcal{D}$, constraint dataset $\mathcal{C}$, initial neural network weights $x_0$
**Require:** Parameters $\mu$, $\eta$, $M_y > 0$, $\tau$, $\beta$, $\rho \geq 0$
1: **for** Iteration $k = 0$ **to** $K - 1$ **do**
2:    Sample $\xi \overset{iid}{\sim} \mathcal{P}_\xi$ and $\zeta_1, \zeta_2 \overset{iid}{\sim} \mathcal{P}_\zeta$
3:    $y_{k+1} = y_k + \eta c(x, \zeta_1)$
4:    **if** $||y_{k+1}|| \geq M_y$ **then**
5:       $y_{k+1} = 0$
6:    **end if**
7:    $x_{k+1} = \text{proj}_\mathcal{X}(x_k - \tau G(x_k, y_{k+1}, z_k; \xi, \zeta_1, \zeta_2))$, where $G$ is defined in (10)
8:    $z_{k+1} = z_k + \beta(x_k - z_k)$
9: **end for**

---

**Algorithm 3** Stochastic Switching Subgradient Method

---

**Require:** Training dataset $\mathcal{D}$, constraint dataset $\mathcal{C}$, initial neural network weights $x_0 \in \mathcal{X}$
**Require:** Total number of iterations $K$, sequence of tolerances of infeasibility $\epsilon_k \geq 0$, sequences of stepsizes $\eta_k^f$ and $\eta_k^c$, mini-batch size $J$, starting index $k_0$ for recording outputs, $I = \emptyset$
1: **for** Iteration $k = 0$ **to** $K - 1$ **do**
2:    Sample a mini-batch $\{\zeta_j\}_{j=1}^J$ so that $\zeta_1, \dots, \zeta_J \overset{iid}{\sim} \mathcal{P}_\zeta$
3:    Set $\bar{c}^J(x_k) = \frac{1}{J}\sum_{j=1}^J c(x_k, \zeta_j)$
4:    **if** $\bar{c}^J(x_k) \leq \epsilon_k$ **then**
5:       Sample $\xi \overset{iid}{\sim} \mathcal{P}_\xi$ and generate $S^f(x_k, \xi)$
6:       Set $x_{k+1} = \text{proj}_\mathcal{X}(x_k - \eta_k^f S^f(x_k, \xi))$ and, if $k \geq k_0$, $I = I \cup \{k\}$
7:    **else**
8:       Sample $\zeta \overset{iid}{\sim} \mathcal{P}_\zeta$ and generate $S^c(x_k, \zeta)$
9:       Set $x_{k+1} = \text{proj}_\mathcal{X}(x_k - \eta_k^c S^c(x_k, \zeta))$ and, if $k \geq k_0$, $I = I \cup \{k\}$
10:    **end if**
11: **end for**
12: **Output:** $x_\tau$ with $\tau$ randomly sampled from $I$ using $P(\tau = k) = \frac{\eta_k}{\sum_{s \in I} \eta_s}$.

---

