# OpenReview forum: "Benchmarking Stochastic Approximation Algorithms for Fairness-Constrained Training of Deep Neural Networks"
_ICLR.cc/2026/Conference — ICLR 2026 Poster_

### Official Review · Reviewer_2stX · 2025-10-28

**Soundness:** 2
**Presentation:** 2
**Contribution:** 2
**Rating:** 2
**Confidence:** 3

**Summary:**

The paper introduces a new benchmark, based on the US census, for testing fairness-constrained learning tasks. The dataset is tested on various methods from the literature, which are implemented in a toolbox.

**Strengths:**

- A standard benchmark specific for testing the fairness of machine learning algorithms can surely be an interesting contribution, and the paper proposes to address this gap

**Weaknesses:**

- While the main claimed contribution of the paper is to provide a benchmark for fairness-constrained learning, I found the structure of the paper unclear, with most of the text reviewing existing literature and not many details on the introduced benchmark and dataset and on what the key novelties and results of the paper are.

- The proposed dataset is an instance of the ACS dataset, already published in [Frances Ding, Moritz Hardt, John Miller, and Ludwig Schmidt. Retiring adult: New datasets for fair machine learning. Advances in Neural Information Processing Systems, 34, 2021], and already used in the context of fairness applications. Consequently, the main contributions of the paper are the implementation and existing comparison of existing algorithms. In my opinion, this contribution, while potentially useful, is too incremental for publication in ICLR

**Questions:**

- What are the main contributions of the paper? Is the proposed dataset an instance of the ACS dataset? What does it make it particularly suitable for fairness testing compared to the original ACS dataset?

---

> ### Author Response · Authors · 2025-11-21
>
> We thank the reviewer for their thoughtful feedback.
> We are pleased that the reviewer found that our benchmark for fairness-constrained deep learning fills a gap in the litterature.
>
> We provide below a point-by-point answer to the reviewer. We have updated the manuscript based on your kind review, **changes in the updated pdf appear in blue**. We would like to emphasize that, based on the reviewers feedback, we have revised the experimental section by (i) improving our hyperparameter search and providing details on the sensitivity of each method  (detailed in Appx B), thus (ii) consolidating the existing experiment on binary attribute (now Sec. 4.1), and (iii) adding a second, multigroup, example (Sec. 4.2).
>
> > "While the main claimed contribution of the paper is to provide a benchmark for fairness-constrained learning, I found the structure of the paper unclear, with most of the text reviewing existing literature and not many details on the introduced benchmark and dataset and on what the key novelties and results of the paper are."
>
> We chose to devote a good portion of the paper to (i) the constrained formulation (4) and its three challenging properties, to (ii) connecting it to the fairness literature so as to have up-to-date fairness notions, and real-world examples, and (iii) to do a thorough literature review of the optimization methods suitable for such a stochastic optimization problem with stochastic constraints.
> Indeed, it is this part of the paper that backs one of the main messages of this paper: there are currently no optimization methods with any convergence guarantee for the fairness-constrained optimization problem. As such, this paper is a call to researchers of the optimization community to develop methods with theoretical guarantees applicable to the described problems.
> The second contribution of the paper is the construction of the toolbox.
> The toolbox is designed to help researchers interested in building novel optimization schemes for such fairness constrained problems to easily compare with existing methods, on real-world problems, with fairness metrics relevant in applications.
> Finally, the toolbox implements as baselines several algorithms for constrained optimization. Our benchmark shows the tradeoffs between these constrained methods and methods from fairness-by-regularization, which may be usefull to practioners already.
>
> > "The proposed dataset is an instance of the ACS dataset"
>
> We stress that we do not propose a dataset, but rather a benchmark for comparison of stochastic optimization algorithms suitable for fairness-constrained machine learning, and conduct an experimental study of the latter. We do use the ACS dataset for training machine learning models in our experiments.
>
> > "Consequently, the main contributions of the paper are the implementation and existing comparison of existing algorithms. In my opinion, this contribution, while potentially useful, is too incremental for publication in ICLR"
>
> Let us stress that, to the best of our knowledge, there was no prior public implementations of the algorithms.
>
>
> > Q1 What are the main contributions of the paper?
>
> The main contributions are
> 1. a literature review of algorithms subject to handling the stochastic optimization under stochastic constraints problem (4), that is grounded in the literature of fairness. Our review highlights that there is currently no algorithm with theoretical guarantees for this problem;
> 2. a toolbox that _i_ implements four algorithms for solving (4), and _ii_ provides an easy-to-use benchmark with real-world fairness constrained problems;
> 3. numerical experiments that compare these algorithms on a real-world dataset, a comparison with alternative approaches to fairness, and a discussion on hyperparameter sensitivity.
>
> > Q2 Is the proposed dataset an instance of the ACS dataset?
>
> As detailed above, we do not propose a dataset, but rather a benchmark for optimization methods. Our work merely uses the ACS dataset to construct learning problems on which we benchmark the optimization methods.
>
> > Q3 What does it make it particularly suitable for fairness testing compared to the original ACS dataset?
>
> Again, we do not propose a dataset but rather a benchmark for optimization methods. We therefore believe that a comparison of our work with the ACS dataset bears little information.
>
> We hope to have addressed all your concerns. We remain at your disposal may you have any further questions or require additional information.

---

### Official Review · Reviewer_S3CZ · 2025-10-28

**Soundness:** 3
**Presentation:** 4
**Contribution:** 3
**Rating:** 6
**Confidence:** 4

**Summary:**

This paper presents a benchmarking study and toolbox for evaluating stochastic approximation algorithms applied to fairness-constrained training of deep neural networks. The authors consider the general constrained empirical risk minimization (ERM) problem, where fairness is imposed through hard constraints (e.g., demographic parity, equal opportunity, equalized odds). They provide an open-source implementation integrating four major stochastic optimization algorithms (Stochastic Ghost (StGh), SSL-ALM, Augmented Lagrangian Method (ALM), and Stochastic Switching Subgradient (SSw)) within a unified framework built on PyTorch and Folktables datasets. The benchmark allows users to automatically construct constrained training formulations and apply fairness constraints across up to 5.7 billion protected subgroups from census-based datasets. Extensive experiments on the ACSIncome dataset compare the convergence speed, fairness violation, and test performance of each algorithm, as well as their robustness under different fairness metrics and constraint formulations. The work aims to standardize empirical evaluation practices in fairness-constrained deep learning and offer a reproducible experimental testbed for future research

**Strengths:**

1- Evaluates four distinct fairness-constrained optimization algorithms under identical experimental setups, providing valuable comparative insights.

2- Offers a transparent, well-engineered implementation with all datasets, hyperparameters, and metrics clearly documented.

3- Bridges fairness theory with realistic deep learning setups, enabling reproducible fairness experiments on real data.

4- Covers three key fairness notions (independence, separation, sufficiency) and links them to optimization constraints.

5- The benchmark includes stochastic ghost gradient methods, augmented Lagrangian, and subgradient-based solvers, giving a broad coverage of optimization paradigms.

**Weaknesses:**

1- Experiments are mainly conducted on a single dataset (ACSIncome), which restricts the scope of empirical validation. Inclusion of more varied domains (e.g., image or language tasks) would strengthen the claim of generality.

2- While results are reported, deeper analysis of when and why certain algorithms perform better (e.g., under which fairness metrics or subgroup imbalances) is missing.

3- Although billions of potential subgroup combinations are mentioned, the experiments do not convincingly demonstrate performance at that scale.

4- The paper does not clearly discuss how the proposed framework will be maintained or integrated with existing fairness toolkits, which may limit its long-term impact.

**Questions:**

1- How does the framework handle multiple simultaneous protected attributes, especially when fairness constraints interact (e.g., intersectional fairness)?

2- Are the results robust to different data distributions or dataset shifts (e.g., subsampled or noisy features)?

3- How computationally expensive are these fairness constraints for large-scale DNNs compared to regularization-based approaches?

4- Could the benchmark include group fairness metrics beyond accuracy, such as calibration or counterfactual fairness?

5- How do hyperparameter settings for fairness constraints (e.g., δ thresholds) influence the convergence behavior of the tested algorithms?

---

> ### Author Response · Authors · 2025-11-21
>
> We thank the reviewer for their thoughtful feedback.
> We are pleased that the reviewer found that this paper bridges fairness theory with realistic deep learning frameworks, provides a valuable comparison between four algorithms that provide a broad coverage of optimization paradigms.
>
> We provide below a point-by-point answer to the reviewer. We have updated the manuscript based on your kind review, **changes in the updated pdf appear in blue**. We would like to emphasize that, based on the reviewers feedback, we have revised the experimental section by (i) improving our hyperparameter search and providing details on the sensitivity of each method  (detailed in Appx B), thus (ii) consolidating the existing experiment on binary attribute (now Sec. 4.1), and (iii) adding a second, multigroup, example (Sec. 4.2).
>
> > W1- Experiments are mainly conducted on a single dataset (ACSIncome), which restricts the scope of empirical validation. Inclusion of more varied domains (e.g., image or language tasks) would strengthen the claim of generality.
>
> While we recognize that we consider learning build from the ACSIncome dataset, we stress that this dataset we employ diverse data from two different states in our two problems, and this dataset is recognized to contain biais and is thus relevant for fairness constrained tasks (FFB: A Fair Fairness Benchmark for In-Processing Group Fairness Methods).
>
> > W2- While results are reported, deeper analysis of when and why certain algorithms perform better (e.g., under which fairness metrics or subgroup imbalances) is missing.
>
> We agree that a study of the impact of different fairness metrics on the performance of constrained optimization algorithms would be useful. However, let us stress that there exist currently no algorithm with a guarantee of convergence in our realistic setup, not to mention a complexity analysis.
> Such a complexity analysis for stochastic programming with stochastic constraints in our realistic set of assumptions is an ambitious research direction.
> The contribution of this paper lies in the identification of this gap in the literature, and the implementation and comparison of different algorithms.
>
> > W3- Although billions of potential subgroup combinations are mentioned, the experiments do not convincingly demonstrate performance at that scale.
>
> Let us stress that we did not claim that our goal was to solve problems with billions of subgroups. We merely mentioned this aspect of the US Census data to hint at its diversity and suitability for building problems with fairness constraints.
> We do agree that many real-world cases are more complex than the setup of the first experiment. Indeed, many combinations of subgroups are possible. In the second experiment (Section 4.2.2) we demonstrate a more complicated setup with a 5-valued protected attribute.
>
> > W4- The paper does not clearly discuss how the proposed framework will be maintained or integrated with existing fairness toolkits, which may limit its long-term impact.
>
> The framework will be maintained on Github and as a Python package with the implemented algorithms and supplementary features. With regards to integration with existing fairness toolkits -- as the framework utilizes PyTorch, in principle, the constraints can be based on any fairness metric implemented in other Python packages, as long as PyTorch autodifferentiation can occur. For example, it is possible to add constraint based on Linear-Fractional Statistics calculated with Fairret.
>
> > Q1- "How does the framework handle multiple simultaneous protected attributes, especially when fairness constraints interact (e.g., intersectional fairness)?"
>
> In the case of multiple protected attributes, one can define subgroups based on combinations of values of protected attributes, and then define a constraint per each subgroup, as formulated in (4). In our view, this approach is consistent with intersectional fairness, as it treats each intersection of protected features as a separate subgroup, and requires some measure of fairness to be achieved between them.
>
> Our second experiment (section 4.2.2) demonstrates a case with multiple constraints; from the optimization point of view, the intersectional case would be similar.
>
> > Q2- "Are the results robust to different data distributions or dataset shifts (e.g., subsampled or noisy features)?"
>
> This is an interesting question indeed.
> A suitable answer would require to quantify the hardness of the stochastic problem, from the subsampling or noise schemes. Nevertheless, we are not aware of any such theoretical developments.
> Nevertheless, we employ in our experiments, learning procedures with a train and test distribution. The plots do show a drift between train and test distributions on the constrained models. As pointed to in the paper, a first theoretical analysis of this matter is (Constrained Learning With Non-Convex Losses, Chamon et al, 2023).

---

> > ### Author Response · Authors · 2025-11-21
> >
> > > Q3- "How computationally expensive are these fairness constraints for large-scale DNNs compared to regularization-based approaches?"
> >
> > Our benchmark shows that the constraint approach is, at least, comparable with the regularization approaches.
> > Indeed, all constrained approaches optimize the loss with efficiency comparable to the regularization baseline (SGD Fairret). Besides, this does not include the computational effort spent in finding suitable hyperparameters. On this front, the sensitivity of regularization methods is well-known in practice, while in our experiments, constrained optimization methods seem to appear less sensitive.
> > Finally, and perhaps most importantly, the constrained approach allows to have a finegrained, and nuanced, control on what should be enforced (the fairness constraints), and what should be optimized (the empirical risk).
> > The situation is much less clear for penalization methods, wherein optimization performs tradeoffs between minimization of empirical risk and of the regularizer on each constraint in an opaque and difficult-to-control way.
> > For more details on this matter, we refer the reviewer to (Juan Ramirez, Meraj Hashemizadeh, and Simon Lacoste-Julien. Position: Adopt Constraints Over Penalties in Deep Learning).
> >
> > > Q4- "Could the benchmark include group fairness metrics beyond accuracy, such as calibration or counterfactual fairness?"
> >
> > In principle, yes, it can. Any fairness metric that qualifies a predictor. Besides, our toolbox is designed to be easily extensible in terms of the fairness metrics.
> >
> > > Q5- "How do hyperparameter settings for fairness constraints (e.g., δ thresholds) influence the convergence behavior of the tested algorithms?"
> >
> > We thank the reviewer for raising this interesting point. We provide ample data on hyperparameter sensitivity of each method, on algorithms parameters, and penalization weight for the regularized method. This is what backs our answer to Question 3.
> > Our purpose in this project was to explore what methods are suitable to solve stochastic optimization with stochastic constraints. We believe that the particular question of quantifying how the difficulty of the optimization problem relates to the hyperparameter $\delta$ is indeed crucial, and will be our next topic of investigation.
> >
> >
> > We hope to have addressed all your concerns. We remain at your disposal may you have any further questions or require additional information.

---

### Official Review · Reviewer_Rxvy · 2025-10-31

**Soundness:** 2
**Presentation:** 3
**Contribution:** 1
**Rating:** 2
**Confidence:** 4

**Summary:**

This paper provides a benchmark of several stochastic optimization algorithms for training deep neural networks under different fairness constraints. They consider datasets built on top of the US Census (Folktables) dataset. They consider the popular group fairness measures independence (statistical parity), sufficiency, and separation. Several experiments have been included in the paper that compare the performance of these existing optimization techniques in minimizing the measures of fairness over the datasets.

**Strengths:**

-- The paper has included a lot of experiments on different algorithmic (stochastic) variants of implementing fairness as a constraint during training. Methods considered include: (i) Stochastic ghost method; (ii) Stochastic smoothed and linearized AL method; (iii) Stochastic switching subgradient method, etc.

-- They consider the three popular fairness notions, and also multiple datasets.

-- Presentation is generally good.

**Weaknesses:**

-- While the vast experiments are highly appreciated, I believe this paper is more suitable as a dataset/benchmark paper. The stochastic optimization algorithms already exist in the literature and have also been used for constraint optimization. The paper applies these constrained optimization variants for the specific constraint of group fairness and studies their performance.

-- Indeed, the paper is quite comprehensive in their experimentation. But, still, the novelty would be limited for such a venue since it is more like a survey of applying different existing techniques to the fairness constraint and seeing the performance. The paper could also be better suited as a survey paper. Though there do exist several other survey papers on fairness in literature, and it would be important to highlight what is the technical gap in existing survey/benchmarking papers that this paper fills.

-- The measures of fairness are mainly the three popular ones.

-- Some works compare tradeoffs between different group fairness measures. E.g. The possibility of fairness: Revisiting the impossibility theorem in practice.
It would be good to compare with them.

**Questions:**

Q1. What would be the gap in existing survey papers or benchmarking papers on algorithmic fairness that this paper fills?

---

> ### Author Response · Authors · 2025-11-21
>
> We thank the reviewer for their thoughtful feedback.
> We are pleased that the reviewer found that our benchmark includes a sufficient number of experiments, that we employ popular fairness notions and multiple datasets, and that our presentation is good.
>
> We provide below a point-by-point answer to the reviewer. We have updated the manuscript based on your kind review, **changes in the updated pdf appear in blue**. We would like to emphasize that, based on the reviewers feedback, we have revised the experimental section by (i) improving our hyperparameter search and providing details on the sensitivity of each method  (detailed in Appx B), thus (ii) consolidating the existing experiment on binary attribute (now Sec. 4.1), and (iii) adding a second, multigroup, example (Sec. 4.2).
>
> > "While the vast experiments are highly appreciated, I believe this paper is more suitable as a dataset/benchmark paper."
>
> We agree, and stress that we have submitted this paper to the "Dataset/Benchmark" primary area of iclr2026.
>
> > "The stochastic optimization algorithms already exist in the literature and have also been used for constraint optimization. The paper applies these constrained optimization variants for the specific constraint of group fairness and studies their performance."
>
> We disagree: there exits no algorithm in the literature for stochastic optimization under *stochastic nonsmooth nonconvex constraints* that have theoretical guarantees. This is one of our contribution: the literature review of optimization methods for that task, outlined in Section 3 and Table 3.
>
> > "Indeed, the paper is quite comprehensive in their experimentation. But, still, the novelty would be limited for such a venue since it is more like a survey of applying different existing techniques to the fairness constraint and seeing the performance. The paper could also be better suited as a survey paper. Though there do exist several other survey papers on fairness in literature, and it would be important to highlight what is the technical gap in existing survey/benchmarking papers that this paper fills."
>
> One of the contributions of this paper is to benchmark algorithms for constrained stochastic training of Machine Learning models; we are not aware of any other paper that shares this goal.
>
> We believe that two novelties of our approach are that (i) we implement three algorithms that handle stochastic programming under restrictive assumptions (linear, weakly convex or smooth constraints), and (ii) we provide a benchmark that compares these methods on real-world settings. This is relevant as it allows (i) practitioners to make an informed choice on which optimization method to use for their fairness constrained tasks (this becomes relevant in Europe with the AI Act that will apply next August), and (ii) researchers to easily compare new algorithm design to the three implemented algorithms, that will act as a baseline.
>
> > The measures of fairness are mainly the three popular ones.
>
> Yes, and the additional Wasserstein distance between prediction distributions.
>
> > Some works compare tradeoffs between different group fairness measures. E.g. The possibility of fairness: Revisiting the impossibility theorem in practice. It would be good to compare with them.
>
> We stress that the primary purpose of this work is not in comparing the fairness measures, but rather the optimization algorithms applied to stochastic optimization problems under stochastic constraints in relation with fairness-related tasks.
> The paper "The possibility of fairness: Revisiting the impossibility theorem in practice" seems to indicate that relaxing mildly the requirements of strict notions of fairness is enough to guarantee the existence of models that meet these formulations. This may have interesting applications on the mathematical properties of the constrained optimization problem (4), in which the parameter $\delta$ needs to be large enough so that the feasible space is not empty. The suggested paper may help in setting the parameter $\delta$. Nevertheless, we believe that exploring these questions theoretically is an ambitious research project, beyond the scope of this benchmarking paper.

---

> > ### Author Response · Authors · 2025-11-21
> >
> > > Q1. What would be the gap in existing survey papers or benchmarking papers on algorithmic fairness that this paper fills?
> >
> > We believe that this paper fills the following gaps in the literature: (i) it provides a review of optimization methods for stochastic programming with stochastic constraints, (ii) it provides an implementation of candidate algorithms for such problems as a python package, (iii) it provides, in the same package, real-world stochastic problems which can be used to easily compare existing algorithms and develop new ones, and (iv) it compares the candidate algorithms in terms of their optimization and fairness performance. We are not aware of existing survey or benchmarking papers with this positionning.
> >
> >
> > We hope to have addressed all your concerns. We remain at your disposal may you have any further questions or require additional information.

---

### Official Review · Reviewer_1Hbo · 2025-11-01

**Soundness:** 3
**Presentation:** 3
**Contribution:** 3
**Rating:** 4
**Confidence:** 4

**Summary:**

The paper presents a benchmark for evaluating stochastic approximation algorithms in fairness-constrained training of deep neural networks. Built on the US Census dataset, the benchmark enables large-scale experiments on fairness objectives formulated as constrained ERM problems. The authors review existing algorithms and implement three recent ones, namely Stochastic Ghost, Stochastic Smoothed and Linearized Augmented Lagrangian Method, and Stochastic Switching Subgradient, and compare them to SGD baselines. The study highlights the lack of unified toolkits and provides a first step toward standardized evaluation for fairness-constrained optimization prooblems.

**Strengths:**

1. This work provides a reproducible and extensible benchmark framework for fairness-constrained deep learning, filling a gap in the literature where no unified platform existed.
2. The writing is very clear, and the notations are consistent. I appreciate Table 3, where the authors review a wide range of stochastic constrained optimization algorithms with a structured taxonomy and theoretical assumptions.
3. The work evaluates multiple fairness criteria, independence, separation, sufficiency, and Wasserstein distance, showing nuanced trade-offs among methods.

**Weaknesses:**

1. The paper primarily implements existing algorithms rather than introducing a new one. While benchmarking is valuable, this may limit perceived theoretical contribution.
2. Only one dataset with a binary protected attribute is used. The scalability and generalization to multiple attributes have not been tested.
3. The presentation of the experimental results can be improved. The current figures are difficult to read.
4. There is no discussion of hyperparameter search across algorithms. Why are the parameters set as in lines 366-375? The results could reflect suboptimal settings rather than intrinsic algorithmic differences.

**Questions:**

1. See some questions in Weaknesses.
2. Could the work extend to multi-group or intersectional attributes?
3. Are there plans to include computational efficiency or memory usage comparisons?

---

> ### Author Response · Authors · 2025-11-21
>
> We thank the reviewer for their thoughtful feedback.
> We are pleased that the reviewer found that our benchmark for fairness-constrained deep learning fills a gap in the literature, that our paper is clear and consistent, and shows multiple fairness criteria and their trade-offs.
>
> We provide below a point-by-point answer to the reviewer. We have updated the manuscript based on your kind review, **changes in the updated pdf appear in blue**. We would like to emphasize that, based on the reviewers feedback, we have revised the experimental section by (i) improving our hyperparameter search and providing details on the sensitivity of each method  (detailed in Appx B), thus (ii) consolidating the existing experiment on binary attribute (now Sec. 4.1), and (iii) adding a second, multigroup, example (Sec. 4.2).
>
> > "The paper primarily implements existing algorithms rather than introducing a new one. [...]"
>
> We disagree: the three algorithms were **not** previously implemented. Indeed, the papers that introduced the algorithms did not provide a github repository to reproduce their results, and we were not able to find any online implementation of the algorithms SSL-ALM, SSW, and Stochastic Ghost online.
>
> > "Only one dataset with a binary protected attribute is used. The scalability and generalization to multiple attributes have not been tested."
>
> We disagree: the ACS dataset features several attributes with more than two values, and we **do** include a use-case with an attribute with 5 values; see Appendix A.
>
> > "The presentation of the experimental results can be improved. The current figures are difficult to read."
>
> Yes, we have updated our figures for improved readability.
>
> > "There is no discussion of hyperparameter search across algorithms. Why are the parameters set as in lines 366-375? The results could reflect suboptimal settings rather than intrinsic algorithmic differences."
>
> We thank the reviewer for raising this interesting issue.
> We have revised the section on experimental results (Section 4) and the appendix (Appx B) with a detailed hyperparameter study. This study shows that [xx].
>
> > "Could the work extend to multi-group or intersectional attributes?"
>
> Yes. In fact, our theoretical formulation already accommodates multi-group and intersectional attributes (see equation 4 and l.156-157). In this framework, each protected subgroup generates a constraint in eq. 4. This handles multi-group attributes directly. Intersectional attributes can be framed as well in eq. 4 by definig the corresponding subgroups and applying the previous approach.
> To better illustrate this point, we now include in Section 4.2 a new experiment with 5 groups setting.
>
> > "Are there plans to include computational efficiency or memory usage comparisons?"
>
> We believe that computational efficiency is best reported by overall runtime, especially since we implement all three algorithms in the same language and with the same effort. This is the philosophy of adopted by some benchmarking efforts in the ML community, such as (Benchopt: Reproducible, efficient and collaborative optimization benchmarks, Massias et al). Overall runtime is the primary measure for the experiments of Section 5. We are happy to further discuss this point.
> We did not discuss memory usage in the document. For first-order methods, such as SSL-ALM, and SSW, the memory cost is proportional to the size of the gradient of the objective plus the sum of the constraints gradients size. The case of Stochastic Ghost is indeed more delicate, as each iteration requires solving several Quadratic Programs.
>
>
> We hope to have addressed all your concerns. We remain at your disposal may you have any further questions or require additional information.

---

### Author Response · Authors · 2025-12-03
**Rebuttal Summary for AC: Reviewer Decisions and Discussion**

Dear AC and reviewers,

We sincerely thank all reviewers and the AC for their time and their constructive comments.
The discussion was cut short before any reviewer could indicate whether our answers satisfied them. It is our hope that we have addressed almost all concerns raised by the reviewers.

Our contributions are summarized below:
- A literature review of three algorithms for solving the constrained training of Deep Neural Networks, as the stochastic-constrained stochastic optimization task outlined in Equation 1 (Reviewer 1Hbo)
- An extensive numerical comparison of these algorithms on a fairness-constrained learning problem (Reviewers 1Hbo, Rxvy, S3CZ)
- The first implementation of these methods in a Python toolkit for constrained training of neural networks, with tools for benchmarking them on real-world fairness-constrained learning tasks and a wide choice of network architectures and fairness constraints (Reviewers 1Hbo, S3CZ, 2stX).

In our rebuttals, we hope to have addressed and clarified the issues posed by the Reviewers. We summarize the main, in our view, points from the discussion below:

- Clarified contribution (Reviewers Rxvy, 2stX): we have clarified the positionning and contribution in the discussion.
- Handling multiple groups (1Hbo): in the revised manuscript, we extend the experiments to feature a protected attribute that admits 5 values (Section 4.2.2). We show that the methods discussed reliably satisfy multiple constraints while minimizing the loss function.
- Added detailed hyperparameter search (Reviewers 1Hbo): in the revised manuscript, we detail and expand our search for hyperparameters for each of the algorithms discussed (Appendix B). We update the section on experimental results (Section 4) accordingly.
- Handling intersectional fairness (Reviewers 1Hbo, S3CZ): the constrained learning formulation (4) can be used to address intersectional fairness by treating each intersection of groups (e.g. White, Married, Male, ...) as an elementary group, which directly fits the setting of the constrained learning formulation (4).
- Extensions for fairness metrics beyond accuracy (Reviewer S3CZ): both the constrained formulation and the benchmark code can be easily modified to replace the loss function $\ell^{s_i}(\theta)$ with a different statistic - for example, to restrict the violation of demographic parity, one would replace it with the Positive Rate. We add here that to restrict the violation of calibration, one would replace it with $P(Y_i=y_i | S_i=s_i, R_i(\theta)=r_i)$, where $R_i(\theta)$ is the risk score predicted by the model.

We have incorporated all changes into the revised manuscript, they appear in blue. We again thank all reviewers and the AC for their valuable time and effort.

Best regards,

The authors

---

### Meta-Review · Area_Chair_8NRy · 2026-01-08

**Summary:**

This paper is a study of stochastic approximation algorithms for fairness-constrained training of deep neural networks. The contributions  include a toolbox that implements four algorithms applicable in real-world scenarios and facilitates benchmarking. They also conduct experiments that compare these algorithms on a real-world dataset.

The concerns raised by the reviewers can be summarised as follows:
1/ While benchmarking is valuable, now new algorithms were proposed.
2/ Generalisation to multiple attributes have not been tested.
3/ No discussion of hyperparameter search.
4/ Experiments are mainly conducted on a single dataset.

**Reviewer Concerns:**

The authors clarified the contributions and addressed the following major concerns:

a/ Handling multiple groups and intersectional fairness (concern 1/). The authors revised the manuscript by extending the experiments, providing additional supporting evidence.
b/ Hyperparameter search (concern 3/). The authors expanded search for each of the algorithm and updated the experimental section of the submission.
c/ Fairness metrics beyond accuracy. The authors extended the submission to address this concern.

In addition the authors provide the first public implementation of the benchmarked algorithm (addressing 1/). They also acknowledged 4/. Arguably, the novelty of this work, submitted to the "Dataset/Benchmark" track, is limited.

**Reviewer Scores:**

Most reviewers voted initially for rejection. The authors provided convincing arguments and additional details to resolve the concerns. While the scope of the study is modest, I would have expected the scores of several reviewers to increase in light of the clarifications provided by the authors making this paper a borderline accept.

---

### Decision · Program_Chairs · 2026-01-26

Accept (Poster)